# INTERNVID: A LARGE-SCALE VIDEO-TEXT DATASET FOR MULTIMODAL UNDERSTANDING AND GENERATION

**Yi Wang**[*,1], **Yinan He**[*,1], **Yizhuo Li**[*,4,1], **Kunchang Li**[6,1], **Jiashuo Yu**[1], **Xin Ma**[3,1]
**Xinhao Li**[2,1], **Guo Chen**[2,1], **Xinyuan Chen**[1], **Yaohui Wang**[1], **Ping Luo**[4,1], **Ziwei Liu**[5,1]
**Yali Wang**[†,6,1], **Limin Wang**[†,2,1], **Yu Qiao**[†,1]
[1]OpenGVLab, Shanghai AI Laboratory   [2]Nanjing University   [3]Monash University
[4]The University of Hong Kong   [5]Nanyang Technological University
[6]Shenzhen Institutes of Advanced Technology, Chinese Academy of Sciences
https://github.com/OpenGVLab/InternVideo/tree/main/Data/InternVid

## ABSTRACT

This paper introduces **InternVid**, a large-scale video-centric multimodal dataset that enables learning powerful and transferable video-text representations for multimodal understanding and generation. **InternVid** contains over 7 million videos lasting nearly 760K hours, yielding 234M video clips accompanied by detailed descriptions of total 4.1B words. Our core contribution is to develop a scalable approach to autonomously build a high-quality video-text dataset with large language models (LLM), thereby showcasing its efficacy in learning video-language representation at scale. Specifically, we utilize a multi-scale approach to generate video-related descriptions. Furthermore, we introduce **ViCLIP**, a video-text representation learning model based on ViT-L. Learned on InternVid via contrastive learning, this model demonstrates leading zero-shot action recognition and competitive video retrieval performance. Beyond basic video understanding tasks like recognition and retrieval, our dataset and model have broad applications. They are particularly beneficial for generating interleaved video-text data for learning a video-centric dialogue system, advancing video-to-text and text-to-video generation research. These proposed resources provide a tool for researchers and practitioners interested in multimodal video understanding and generation.

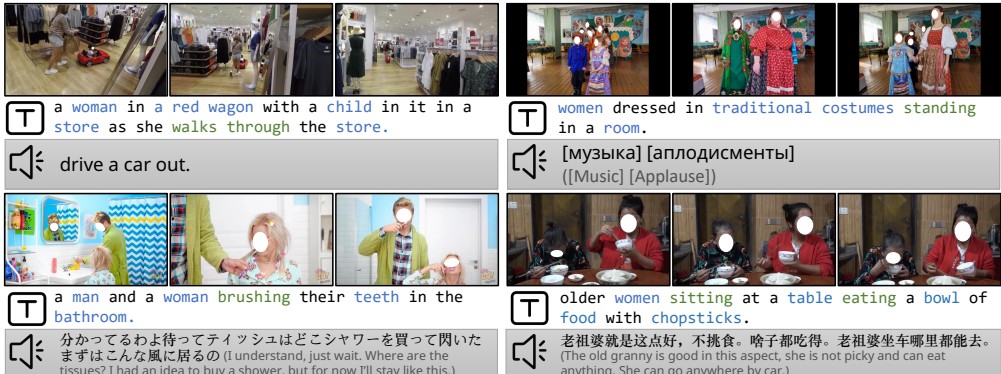

Figure 1: Examples (we give three frames of each video clip), the corresponding generated captions, and ASR transcripts in InternVid. In the captions, we highlight nouns in blue and verbs in green. Non-English transcripts are translated to English using LLM (Brown et al., 2020).

---

* Equal contribution.   † Corresponding authors.

# 1 INTRODUCTION

Learning transferable video-text representations is both challenging and essential for video understanding in various real-world applications, such as autonomous driving, intelligent surveillance, human-computer interaction, to name a few. While contrastive learning using web-scale data has been successful in image-text representation, it remains underexplored in the video-text domain.

A key reason for this limited exploration is *the lack of a high quality video-language dataset for pretraining at scale*. Current research relies on datasets like HowTo100M (Miech et al., 2019), HD-VILA (Xue et al., 2022), and YT-Temporal (Zellers et al., 2021; 2022), whose texts are generated using automatic speech recognition (ASR). Despite their large scale, these datasets often have low semantic correlations between the videos and corresponding textual descriptions (Miech et al., 2019; Xue et al., 2022; Zellers et al., 2021; 2022). Empirical studies demonstrate that improving this correlation (e.g. aligning videos with subtitles to improve their matching) significantly benefits downstream tasks such as video retrieval and video question answering (Bain et al., 2021). Recent works have utilized WebVid10M (Bain et al., 2021), a dataset with higher-quality alt-texts, to address the low video-text correlation issue. However, its limited scale and dynamics hinder its use in current data and model scaling studies. Specifically, only 10M video-text pairs are provided, and the depicted scenes contain relatively few actions or activities.

We propose a large-scale video-centric dataset InternVid to address the challenge of *scaling up video-language modeling while maintaining high video-text correspondence*. Visual examples are given in Figure 1. Note the ASR transcripts barely depict visual elements in videos while the generated captions do. The dataset contains highly-correlated video-text pairs and includes over *7 million videos*, totaling *760,000 hours* and resulting in *234 million video clips*, with various subsets for different needs. These videos cover *16 scenarios* and around *6 thousand motion descriptions*. To improve video-text matching, we generate captions using a multiscale approach. In the coarse scale, we caption the middle frame of each video and use the description as the video caption. In the fine scale, we produce and summarize frame-by-frame captions with a language model.

Leveraging InternVid, we scale a video-language transformer (ViT-L) in contrastive learning from a data perspective, and its experiments prove InternVid enables learning scalable video-text models. We introduce video masking to the model to accelerate the whole learning without compromising its effectiveness. The video and text encoders are initialized from the CLIP pretrained model with the same scale. With InternVid, we learn a video-text model for several epochs, achieving impressive zero-shot performance. Compared with previous Video CLIP variants, our proposed ViCLIP shows notable performance improvement, especially in zero-shot settings.

In addition to large-scale video-language contrastive pretraining, we discover its effectiveness in producing interleaved video-text data for learning a video-centric dialogue system like Flamingo (Alayrac et al., 2022; Awadalla et al., 2023), and advancing video generation. Since the text-annotated clips are extracted from videos, we naturally collect clips and their corresponding text based on the sampling locations. This results in approximately 7 million interleaved data pieces, suitable for instruction tuning as multi-turn video-centric dialogue. For video generation, we filter the core set and obtain 18 million video clips. Alongside WebVid-10M, InternVid can significantly improve a stable-diffusion based video generation model to new heights.

In summary, our contributions are threefold.

- We introduce a new web-scale video-language dataset InternVid. This dataset, aimed at advancing video-related multimodal understanding and generation at scale, is created using a multi-scale video captioning approach powered by LLM, ensuring high-quality video-text data with minimal human intervention. It includes computational features (video-text correlation and visual aesthetics) across the whole dataset and gives way to diverse subsets to cater to varying training needs.

- We learn a new video-language model, ViCLIP, which is trained on InternVid using ViT-L. It incorporates both contrastive learning and mask modeling, allowing for efficient learning of transferrable video-language representation. This model achieves state-of-the-art zero-shot action recognition in Kinetics, scoring 75.7, 73.5, and 66.4 on K400, K600, and K700 with the average top1 and top5 accuracies, respectively. It gets competitive performance on video retrieval, setting a new baseline for video-text understanding.

- InternVid fosters the development of multimodal dialogue systems and text-to-video generation. ViCLIP learned on InternVid could serve as a backbone of video-centric dialogue systems (Zhu et al., 2023a; Li et al., 2023c; Liu et al., 2023), conducting tasks as action recognition, temporal

| Dataset | Caption | Domain | #Videos | #Clips | Len$_{Clip}$ | Len$_{Cap}$ | Dur(h) | Res |
|---|---|---|---|---|---|---|---|---|
| MSR-VTT (Xu et al., 2016) | Manual | open | 7.2K | 10K | 15.0 | 9.3 | 40 | 240P |
| DideMo (Anne Hendricks et al., 2017) | Manual | Flickr | 10.5K | 27K | 6.9 | 8.0 | 87 | - |
| LSMDC (Rohrbach et al., 2017) | Manual | movie | 200 | 118K | 4.8 | 7.0 | 158 | 1080P |
| YouCook2 (Zhou et al., 2018) | Manual | cooking | 2K | 14K | 19.6 | 8.8 | 176 | - |
| How2 (Sanabria et al., 2018) | Manual | instruct | 13.2K | 80K | 90.0 | 20.0 | 2K | - |
| ANet Caption (Krishna et al., 2017) | Manual | action | 20K | 100K | 36.0 | 13.5 | 849 | - |
| VideoCC3M (Nagrani et al., 2022) | Transfer | open | 6.3M | 10.3M | 10 | - | 17.5K | - |
| WebVid10M (Bain et al., 2021) | Alt-text | open | 10.7M | 10.7M | 18.0 | 12.0 | 52K | 360P |
| WTS70M (Stroud et al., 2020) | Metadata | action | 70M | 70M | 10 | - | 194K | - |
| HowTo100M (Miech et al., 2019) | ASR | instruct | 1.2M | 136M | 3.6 | 4.0 | 134.5K | 240P |
| HD-VILA-100M (Xue et al., 2022) | ASR | open | 3.3M | 103M | 13.4 | 32.5 | 371.5K | 720P |
| YT-Temporal-180M (Zellers et al., 2021) | ASR | open | 6M | 180M | - | - | - | - |
| InternVid (ours) | Generated | open | 7.1M | 234M | 11.7 | 17.6 | 760.3K | 720P |

Table 1: Statistics of InternVid and its comparison with existing video-language datasets.

understanding, reasoning, and creativity within an open-ended environment. Furthermore, we provide a subset, InternVid-Aes, aiding in generating high-resolution watermark-free videos. Utilizing InternVid-Aes, both visual and quantitative outcomes of a text-to-video baseline can be noticeably enhanced (FVD: 705.3 → 616.5).

# 2   RELATED WORK

**Multimodal Datasets.** Vision-text data are necessary to enable crossmodal learning. To learn vison-language representation effectively, these datasets should be large at scale and high at vision-text correlations. To this end, researches usually leverage existing web images with alt-text (Thomee et al., 2016; Sharma et al., 2018; Changpinyo et al., 2021; Hu et al., 2022; Desai et al., 2021; Schuhmann et al., 2022) and videos with ASR transcriptions (Miech et al., 2019; Zellers et al., 2021; 2022; Xue et al., 2022; Bain et al., 2021; Srinivasan et al., 2021) for scalable learning.

For video-centric multimodal datasets, HowTo100M (Miech et al., 2019) collected instructional YouTube videos and exploited the corresponding ASR subtitles for learning joint representations. Nagrani et al. (2022) proposed VideoCC3M by transferring image-text datasets to video ones. Zellers et al. (2021; 2022) and Xue et al. (2022) proposed YT-Temporal and HD-VILA for Audio-Visual-Text joint learning and high-resolution video crossmodal learning, respectively. On the other hand, Bain et al. (2021) found video-text alignment matters more than their quantities, so they produced WebVid (Bain et al., 2021) where 10M videos with the corresponding alt-texts. This is frequently employed in recent video-language pretraining approaches (Li et al., 2023d; Cheng et al., 2023).

**Video Understanding.** Pretraining large-scale video-text models and fine-tuning them for downstream tasks has become the norm in the video-language field (Miech et al., 2020; Li & Wang, 2020; Xu et al., 2021; Li et al., 2023d; 2022a; Xu et al., 2021; Hu et al., 2022; Dou et al., 2022; Shen et al., 2021; Yao et al., 2021; Sun et al., 2019; Zhu & Yang, 2020; Wang et al., 2022b; Chen et al., 2022; Zellers et al., 2021; 2022; Zeng et al., 2023b;a; Chen et al., 2023a;b; He et al., 2023; Chen et al., 2023c). Early techniques (Sun et al., 2019; Zhu & Yang, 2020) used pretrained visual and language encoders to obtain offline video and text features, but recent methods (Li & Wang, 2020; Miech et al., 2020; Hu et al., 2022; Dou et al., 2022; Tong et al., 2022; Wang et al., 2023) highlight the advantages of end-to-end training. Common practices include two or three pretraining tasks, such as masked language modeling (Li et al., 2022b), video-text matching (Wang et al., 2022a), video-text contrastive learning (Xu et al., 2021; Wang et al., 2022b), masked video modeling (Tong et al., 2022; Wang et al., 2023; 2022b), and video-text masked modeling (Fu et al., 2021).

In the multimodal video context, VIOLET (Fu et al., 2021) combined masked language and video modeling, while All-in-one (Wang et al., 2022a) proposed a unified pretraining with a shared backbone. LAVENDER (Li et al., 2022b) unified tasks through masked language modeling. Despite their success in multimodal benchmarks, these methods' reliance on limited video-text data hampers performance in video-only tasks like action recognition. Conversely, InternVideo (Wang et al., 2022b) and UMT (Li et al., 2023d) combined masked modeling with crossmodal contrastive learning, leading to competitive performance in both video-only and video-language tasks. MERLOT Reserve (Zellers et al., 2022) exploited 20 million video-text-audio pairs for training joint video representations using contrastive matching, setting new standards in video recognition and visual commonsense reasoning. To address modality entanglement in crossmodal learning, mPLUG-2 (Xu et al., 2023) introduced a shared module across image, video, and text to encourage modality collaboration while reserving modality-specific modules for their differences.

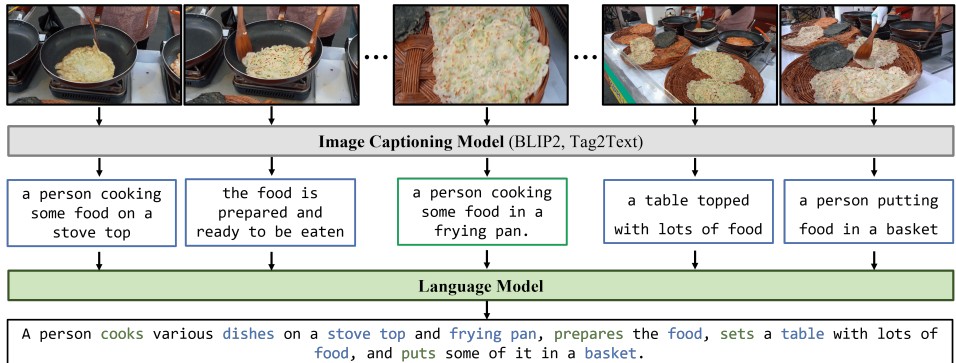

Figure 2: The proposed multiscale video caption pipeline. The captions in coarse and fine scales are marked in green and blue, respectively.

# 3 INTERNVID: A VIDEO-CENTRIC MULTIMODAL DATASET

A high-quality video-text dataset at scale is a premise to conduct large-scale video-language learning and associated tasks. We build this dataset considering three factors: substantial temporal dynamics, rich and diverse semantics, and strong video-text correlations. To improve the temporal dynamics in the dataset, we gather videos retrieved using action/activity-based query words. For rich and varied semantics, we not only crawl trending videos across various categories but also deliberately increase the proportion of data consciously collected from various countries and languages. To strengthen video-text correlations, we employ image captioning and language models to generate video descriptions from frame-specific annotations. Next, we elaborate the dataset construction process and discuss its characteristics.

## 3.1 DATA CURATION

We collect videos from YouTube considering the diversity and richness of its data, and its support for academic usage. We obtain 7 million public YouTube videos with an average duration of 6.4 minutes, covering 16 topics. To avoid the overlap with the existing datasets, we acquire videos by creating a database of YouTube video IDs and excluding any videos already present in publicly available datasets (released prior to April 2023). The data curation strategies are two-fold. On one hand, We select popular channels and the corresponding hot or high-rated videos from the categories e.g. news, gaming, etc., resulting in 2 million videos. On the other hand, we create a list of motion descriptions. With it, we obtain 5.1 million videos by choosing the top retrieved ones.

**Defining Actions for Queries.** We define around 6K action phrases from American Time Use Survey (ATUS), public video datasets, and text corpus. Then they are refined both manually and automatically. We employ actions from ATUS from 2017 to 2022 (Heilbron et al., 2015), merging them and removing the duplicates. For the used video datasets, we leverage Kinetics (Carreira & Zisserman, 2017), SomethingSomething (Goyal et al., 2017; Mahdisoltani et al., 2018), UCF101 (Soomro et al., 2012), and so on. This provides us with 1103 action labels. Moreover, we access several visual grounding corpus (Song et al., 2021; Yang et al., 2022; Li et al., 2017). A language model (Brown et al., 2020) is employed to extract actions and their corresponding targets (if exist) to form phrases from the corpus, leading to 5001 actions with manual checking. Totally, we collect 6104 action queries for searching videos.

**Collection Strategies.** For quality control, we establish crawling rules. We collect videos that are between 10 seconds and 30 minutes in duration and have resolutions ranging from 360P to 720P. Videos with resolutions above 720P are processed to 720P. To enrich the dataset descriptions, we gather videos with their audio, subtitles, titles, summaries, and other metadata.

**Trimming.** We segment videos into clips using scene variance. We directly employ the corresponding filter in PySceneDetect with a threshold as 27. During this procedure, we also filter out clips in still or extreme dynamics (e.g. a browse of a photo gallery). After the filtering, we get 234M clips.

## 3.2 MULTISCALE VIDEO CAPTIONING

To generate video captions that are scalable, rich, and diverse, we employ a multiscale method with two distinct captioning strategies, as depicted in Figure 2. On the finer scale, we simplify the video captioning process by concentrating on the common objects, actions, and scene descriptions within the video clip. We deliberately overlook intricate details such as subtle facial expressions & movements, and other nuanced elements. On the coarser scale, we adopt the single-frame bias assumption from

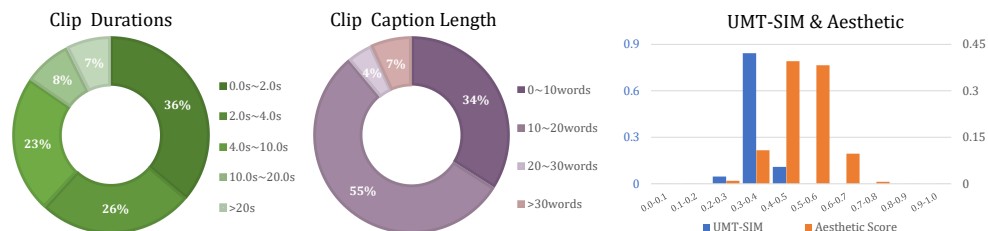

Figure 3: Video statistics in InternVid. It encompasses a diverse set of categories, gathered from multiple countries and averaging a duration of five minutes.

Figure 4: Clip statistics in InternVid. InternVid contains a diverse distribution of clip durations and caption lengths. It also offers aesthetic scores and multimodal similarity scores for each clip.

(Lei et al., 2022) and exclusively caption the central frame of the video. Given our focus on brief clips (around 10 seconds) filtered via scene segmentation, most videos predominantly display consistent objects without substantial appearance alterations. This circumvents the identity-preserving issue when dealing with videos from image perspectives. We employ the lightweight image captioning model Tag2Text (Huang et al., 2023b) for the finer scale, which describes videos at low fps in a frame-by-frame manner. These individual image captions are then synthesized into a comprehensive video description using a pretrained language model T5-summary (Raffel et al., 2020; Chiang et al., 2023). At the coarser scale, we use BLIP2 (Li et al., 2023b) to caption the middle frame of the clip.

### 3.3 STATISTICS AND FEATURES

We present the statistics of InternVid with other video-language datasets in Table 1. We collected videos from 16 mainstream categories, as in Figure 3. Unlike prior studies (Miech et al., 2019; Xue et al., 2022; Zellers et al., 2021), we emphasize diversity by selecting videos from countries with different languages instead of only English. In terms of duration, every video lasts 351.9s on average. Almost half (49%) of the videos are five minutes or less, while a quarter (26%) fall between five and ten minutes. Among the curated videos, 85% were high-resolution (720P), while the remaining 15% had lower resolutions ranging from 360P to 720P. Though the lower-resolution videos may not support high-quality video generation, they can still be useful in video-text representation learning.

InternVid exhibits diverse clip durations and caption lengths at the clip level. The aesthetic scores and clip-caption similarities are distributed uniformly, as shown in Figure 4. The majority of clips are 0-10 seconds in length, accounting for 85% of all clips (Figure 4: left). Approximately half of the clips have captions with 10-20 words. The statistics of the captions and transcripts is given in App. C.

We measured the aesthetic scores and clip-caption similarity of all clips using the models in (Schuhmann et al., 2022) and (Li et al., 2023d), respectively, as given in Figure 4: right. Based on these scores, we can build different versions of InternVid for various purposes. We uniformly sampled four frames of each clip, calculated their aesthetic scores, and took the maximum score as the video aesthetic score. For clip-caption similarity, we computed the cosine similarity between video embeddings and text ones, again using a uniform sampling of four frames for each clip. Most clips score around 4-6 in aesthetics, accounting for approximately 75% of the data. For UMT-SIM [1], over 80% of the clips scored between 0.3-0.4.

### 3.4 INTERLEAVED VIDEO-TEXT DATA GENERATION

Utilizing video-caption data, we can develop an interleaved video-text dataset for generative multimodal pretraining. Previous researches (Alayrac et al., 2022; Awadalla et al., 2023; Huang et al., 2023a; Zhu et al., 2023b) confirm that pretraining on the interleaved image-text sequences results in significant multimodal in-context abilities. Our work makes the initial step in creating a large-scale

---

[1] UMT-SIM refers to the use of Unmasked Teacher (UMT) (Li et al., 2023d) to compute the similarity score between a given video clip and the accompanying text.

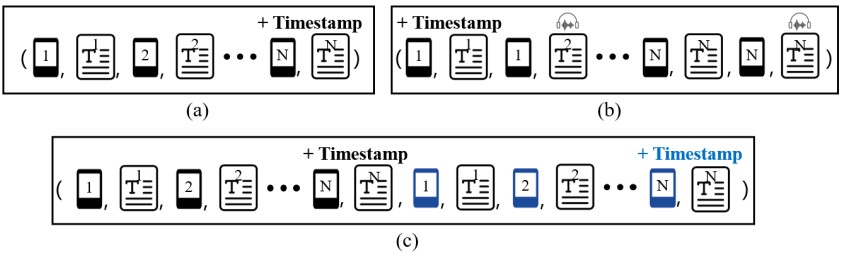

Figure 5: Interleaved video-text data generation in InternVid with three formats.

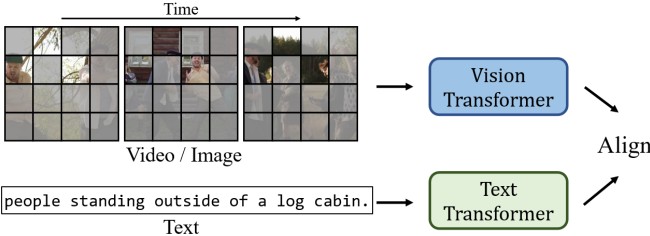

Figure 6: Framework of ViCLIP.

interleaved video-text dataset InternVid-ICL, containing 7.1M interleaved video-text data pairs. We give three methods for organizing clips and their captions: (1): Arrange clips and their descriptions sequentially based on their temporal order within the same video, as illustrated in Figure 5 (a). (2): Enhance diversity in interleaved video-text items by assigning ASR text to a used clip in addition to its caption, as demonstrated in Figure 5 (b). (3): Extend method (1) by concatenating two interleaved multimodal items, creating a video-centric dialogue simulating user queries involving multiple videos (Figure 5 (c)). One visual example of these arrangements is provided in Table 8.

## 4  VICLIP: LEARNING VIDEO-TEXT REPRESENTATION AT SCALE

Built upon CLIP (Radford et al., 2021), we make a simple video-text pretraining baseline ViCLIP. It consists of a video encoder (ViT) (Dosovitskiy et al., 2021) and a text encoder, as given in Figure 6. Both modules are initialized from the corresponding CLIP components. We update the native attention in the video encoder to spatiotemporal attention while maintaining other designs. For efficient learning, we apply masking to videos in pre-training. The optimization target is the contrastive loss between input video and text embeddings.

**Video & Text Encoders with Masking Learning.** Our video encoder uses a ViT with spatiotemporal attention. We apply random patch masking following MAE-based methods (Tong et al., 2022; Wang et al., 2023) to the input videos. It significantly alleviates the computational burden. The used text encoder is also a transformer followed by (Radford et al., 2021; Schuhmann et al., 2022).

**Unmasked Video-Text Pretraining.** We feed all visual tokens into the video transformer instead of just the masked ones towards the end of the pretraining process. This helps bridge the gap between pretraining and downstream applications where the full video is used as input. We perform unmasked training for 0.5 epochs with a learning rate of 4e-6.

**Training Objectives.** Our framework optimizes video-text alignment. It minimizes InfoNCE loss (Oord et al., 2018) using global video and text features, as

$$\mathcal{L}_C = \mathcal{L}_C^{\mathbf{V}\to\mathbf{T}} + \mathcal{L}_C^{\mathbf{T}\to\mathbf{V}} = -\sum_{i=1}^{N} \log \frac{\exp(\mathrm{sim}(f_i^{\mathbf{V}}, f_i^{\mathbf{T}})/\tau)}{\sum_{j=1}^{N} \exp(\mathrm{sim}(f_i^{\mathbf{V}}, f_j^{\mathbf{T}})/\tau)} - \sum_{i=1}^{N} \log \frac{\exp(\mathrm{sim}(f_i^{\mathbf{T}}, f_i^{\mathbf{V}})/\tau)}{\sum_{j=1}^{N} \exp(\mathrm{sim}(f_i^{\mathbf{T}}, f_j^{\mathbf{V}})/\tau)}, \quad (1)$$

where $f^{\mathbf{V}}$ and $f^{\mathbf{T}}$ denote the learned video and text embeddings, respectively. $\mathrm{sim}(\cdot)$ computes the cosine similarity between two features. $\tau$ is the learnable temperature.

## 5  EXPERIMENTS

We learn ViCLIP on subsets of InternVid and evaluated its performance on video-related benchmarks using full-finetuned and zero-shot settings. We sample subsets InternVid-10M, InternVid-50M, and InternVid-200M randomly. Sampling details are given in App. E.1.

| Method | Training Data | K400 | | K600 | | K700 | |
|---|---|---|---|---|---|---|---|
| | | top-1 (↑) | AVG (↑) | top-1 (↑) | AVG (↑) | top-1 (↑) | AVG (↑) |
| CLIP (Radford et al., 2021) | CLIP400M (Radford et al., 2021) | 58.42 | 70.14 | 55.11 | 67.16 | 46.12 | 58.38 |
| CLIP (Radford et al., 2021) | DataComp-1B (Gadre et al., 2023) | 56.14 | 67.67 | 54.15 | 65.83 | 45.36 | 57.01 |
| EVA-CLIP-L (Sun et al., 2023) | Merged-2B (Sun et al., 2023) | - | 65.00 | - | 64.90 | - | 59.10 |
| EVA-CLIP-E (Sun et al., 2023) | LAION-2B (Schuhmann et al., 2022) | - | 69.80 | - | 69.30 | - | 63.40 |
| ViCLIP | +WebVid10M (Bain et al., 2021) | 59.88 | 71.03 | 58.66 | 69.84 | 50.23 | 61.86 |
| ViCLIP | +InternVid-10M | 56.68 | 68.17 | 54.67 | 66.28 | 46.53 | 58.73 |
| ViCLIP | +InternVid-50M | 57.18 | 68.93 | 55.36 | 67.07 | 47.00 | 59.36 |
| ViCLIP | +InternVid-200M | 59.80 | 71.09 | 57.80 | 69.34 | 49.30 | 61.25 |
| ViCLIP | +InternVid-10M-DIV | 63.00 | 74.15 | 60.68 | 72.07 | 52.50 | 64.59 |
| ViCLIP | +InternVid-10M-FLT | **64.80** | **75.70** | **62.20** | **73.53** | **54.30** | **66.38** |

Table 2: Zero-shot action recognition results on Kinetics 400/600/700.

| Method | Training Data | K400 | | SthSthV2 | |
|---|---|---|---|---|---|
| | | top-1 (↑) | top-5 (↑) | top-1 (↑) | top-5 (↑) |
| CLIP (Radford et al., 2021) | CLIP400M (Radford et al., 2021) | 86.7 | 97.2 | 70.1 | 92.5 |
| CLIP (Radford et al., 2021) | DataComp-1B (Gadre et al., 2023) | 85.6 | 96.8 | 68.9 | 91.8 |
| ViCLIP | +WebVid10M (Bain et al., 2021) | 85.0 | 96.8 | 68.7 | 91.9 |
| ViCLIP | +InternVid-10M-FLT | 86.8 | 97.5 | 71.2 | 93.2 |
| ViCLIP | +InternVid-10M-FLT+K710 | 88.0 | 97.8 | 71.8 | 93.6 |
| ViCLIP | +InternVid-200M | 87.9 | 97.9 | 73.6 | 94.9 |
| ViCLIP | +InternVid-200M+K710 | **88.7** | **98.2** | **74.2** | **95.0** |

Table 3: Fine-tuned action recognition results on Kinetics 400 and SomethingSomethingV2.

## 5.1 TRANSFERABLE VIDEO REPRESENTATION PERFORMANCE

**Action Recognition.** In addition to OpenAI's CLIP-L (CLIP400M (Radford et al., 2021)) and LAION (DataComp-1B (Gadre et al., 2023)), we also include EVA-CLIP-L/14 and EVA-CLIP-E/14 (Sun et al., 2023) for comparison. More experimental settings are given in App. E.2.

*Zero-Shot.* Table 2 shows that when trained on InternVid-10M-FLT, ViCLIP outperforms all other methods, including EVA-CLIP-E. This result validates InternVid's effectiveness in learning video-text embeddings. Note that ViCLIP with InternVid-10M-FLT sets new records on zero-shot action recognition in Kinetics 400/600/700, demonstrating a significant performance boost compared to ViCLIP with WebVid10M or other models. Moreover, ViCLIP trained on InternVid-10M-FLT exceeds its performance on InternVid-200M. Normally, we would expect the model trained on InternVid-200M to perform better than those on -10M-DIV or -FLT, given that the latter two subsets derive from the former. Unless this discrepancy results from improper learning, we conjecture that false negative samples could severely impede video-text contrastive learning if we don't purposefully reduce the number of clips taken from the same video. Specifically, we hypothesize that clips from the same video share similar representations and captions. Contrastive learning, however, assumes these clips to be different. This situation also undermines the significance of using a large batch size in current training since it increases the probability of encountering more false negatives.

*Fine-tuned.* In Table 3, note when comparing ViCLIP trained on InternVid with image CLIP models or ViCLIP trained with WebVid, there is a clear increase in accuracy. Unlike the zero-shot results, when ViCLIP is pretrained with a larger number (200M) of video-text data pairs, it achieves higher accuracy in fine-tuned recognition tasks (87.9% in K400 and 73.6% in SthSthV2) compared to when pretrained (86.8% in K400 and 71.2% in SthSthV2) with fewer data (10M). This suggests that InternVid provides greater benefits for fine-tuned action-related tasks. The decrease in performance of ViCLIP with WebVid highlights the importance of addressing the distribution gap between WebVid and the action videos used for evaluation, emphasizing the need to collect videos with evident temporal dynamics.

**Video-Text Retrieval.** We evaluate the video retrieval performance of baselines and ViCLIP using different pretraining datasets on five popular benchmarks (Heilbron et al., 2015; Xu et al., 2016; Rohrbach et al., 2015; Anne Hendricks et al., 2017; Chen & Dolan, 2011), as shown in Table 4 and 5. We uniformly sample eight frames from the input videos. For the CLIP models from OpenAI (Radford et al., 2021) and LAION (Schuhmann et al., 2022), we utilize their officially released ViT-L models and extract video embeddings by averaging the computed frame-wise image embeddings. Our ViCLIP directly predicts video embeddings. For evaluating retrieval performance, *we report R@1 scores for both text-to-video (t2v) and video-to-text (v2t) tasks* in 4 and 5.

Both Table 4 and 5 demonstrate that video-language pretraining is crucial for enhancing fine-tuned and zero-shot retrieval performance. This point is substantiated by the comparison between CLIP and ViCLIP using InternVid-50M. Table 4 exhibits a boost of nearly 4-10 points across different benchmarks in the zero-shot setting. Meanwhile, Table 5 shows an increase of approximately 10 points across all R@1 scores in the fine-tuned setting.

| Method | Data | MSR-VTT | | LSMDC | | DiDeMo | | MSVD | | ANet | |
|---|---|---|---|---|---|---|---|---|---|---|---|
| | | T2V | V2T | T2V | V2T | T2V | V2T | T2V | V2T | T2V | V2T |
| CLIP (Radford et al., 2021) | CLIP400M (Radford et al., 2021) | 29.0 | 25.8 | 13.9 | 15.2 | 11.5 | 19.1 | 37.9 | 60.0 | 8.3 | 12.2 |
| CLIP (Radford et al., 2021) | DataComp-1B (Gadre et al., 2023) | 30.4 | 24.2 | 13.9 | 11.9 | 12.7 | 18.7 | 40.5 | 57.2 | 9.1 | 13.2 |
| CLIP4Clip (Luo et al., 2022) | +HowTo100M (Miech et al., 2019) | 32.0 | - | 15.1 | - | - | - | 38.5 | - | - | - |
| ViCLIP | +WebVid10M (Bain et al., 2021) | 35.6 | 33.1 | 16.5 | 13.4 | 14.5 | 23.3 | 45.3 | 69.0 | 12.4 | 19.0 |
| ViCLIP | +InternVid-10M | 36.4 | 37.1 | 17.1 | 15.0 | 16.4 | 25.9 | 45.2 | 69.8 | 13.5 | 23.4 |
| ViCLIP | +InternVid-50M | 39.7 | 40.7 | 18.0 | 16.7 | 16.7 | 26.4 | 46.5 | 72.2 | 13.6 | 23.2 |
| ViCLIP | +InternVid-200M | 39.3 | 39.5 | 18.3 | 16.6 | 17.1 | 25.5 | 47.3 | 70.0 | 13.7 | 21.6 |
| ViCLIP | +InternVid-10M-DIV | 41.5 | **41.6** | 18.5 | **17.4** | 17.7 | 26.2 | 48.6 | 71.9 | 14.8 | 23.4 |
| ViCLIP | +InternVid-10M-FLT | **42.4** | 41.3 | **20.1** | 16.9 | **18.4** | **27.9** | 49.1 | 75.1 | 15.1 | 24.0 |

Table 4: Results of zero-shot video retrieval on MSR-VTT, LSMDC, DiDeMo, MSVD, and ANet.

| Method | Data | MSR-VTT | | LSMDC | | DiDeMo | | MSVD | | ANet | |
|---|---|---|---|---|---|---|---|---|---|---|---|
| | | T2V | V2T | T2V | V2T | T2V | V2T | T2V | V2T | T2V | V2T |
| CLIP (Radford et al., 2021) | CLIP400M (Radford et al., 2021) | 38.2 | 38.7 | 22.5 | 22.6 | 32.2 | 33.9 | 67.3 | 69.9 | 26.1 | 26.9 |
| CLIP (Radford et al., 2021) | DataComp-1B (Gadre et al., 2023) | 37.2 | 37.5 | 18.7 | 18.5 | 33.5 | 34.2 | 66.3 | 70.2 | 24.5 | 25.8 |
| CLIP4Clip (Luo et al., 2022) | +HowTo100M Miech et al. (2019) | 45.6 | 45.9 | 24.3 | 23.8 | 43.0 | 43.6 | 45.2 | 48.4 | 40.3 | 41.6 |
| ViCLIP | +WebVid10M (Bain et al., 2021) | 50.8 | 49.3 | 27.3 | 28.4 | 48.1 | 48.5 | 76.7 | **81.2** | 44.5 | 43.2 |
| ViCLIP | +InternVid-10M | 51.8 | 49.7 | 28.5 | 29.4 | 49.5 | 50.6 | 77.2 | **80.0** | 49.7 | 48.4 |
| ViCLIP | +InternVid-50M | 52.8 | 52.2 | 30.9 | 30.9 | 49.4 | 48.7 | 78.1 | **80.0** | 49.7 | 49.0 |
| ViCLIP | +InternVid-200M | 53.7 | **53.4** | 29.3 | 31.3 | 51.1 | 50.8 | **79.9** | 78.4 | **52.8** | **51.1** |
| ViCLIP | +InternVid-10M-DIV | **55.0** | 53.3 | 32.0 | 30.0 | **51.7** | **52.1** | 75.8 | 77.8 | 50.4 | 48.9 |
| ViCLIP | +InternVid-10M-FLT | 52.5 | 51.8 | **33.0** | **32.5** | 49.4 | 50.2 | 77.2 | 79.0 | 49.8 | 48.1 |

Table 5: Results of fine-tuned video retrieval on MSR-VTT, LSMDC, DiDeMo, MSVD, and ANet.

*Zero-Shot.* Table 4 reveals InternVid-10M outperforms WebVid when employing the same method, ViCLIP, with an average increase of 6.3% in R@1 across nearly all benchmarks. This improvement can be further amplified by diversifying the training clips used, as InternVid-10M-DIV and -FLT surpass WebVid on ViCLIP with gains in R@1 of 14.0% and 17.1%, respectively. These results underline, once again, the effectiveness of the correspondence between our generated video captions and their corresponding videos. Comparing CLIP4Clip using HowTo100M with ViCLIP using Web-Vid10M or InternVid-10M shows that the correlation between video and text influences performance more significantly than their quantity. Moreover, the zero-shot performance demonstrates that the video-text representation learned using InternVid is transferable. This claim is supported by its superior performance across multiple video retrieval benchmarks.

*Fine-Tuned.* Table 5 exhibits a noticeable improvement when transitioning from InternVid-10M to WebVid10M while using ViCLIP for both t2v and v2t retrieval across almost all datasets. On average, there is a 3.7% increase in t2v R@1 across all benchmarks, with particularly significant rise observed in ActivityNet (an increase of over 11.9%). However, ViCLIP using WebVid10M yields better v2t R@1 scores than when using InternVid-10M (81.2 vs. 80.0). We believe this does not alter the overall trend that InternVid-10M generally provides more advantage to ViCLIP than WebVid10M does.

The benefits of used video data become even more apparent when comparing InternVid-10M-DIV or InternVid-10M-FLT with WebVid10M. Their overall increases are 5.8% and 5.1%, respectively. Despite these improvements, issues related to data diversity persist.

**Data Scaling and Issues.** Figure 7 and 8 illustrate how ViCLIP's performance changes in zero-shot and fine-tuning settings when varying the scale of InternVid. In both scenarios, increasing the data scale results in significant increases in performance. As shown in Figure 7, ViCLIP's discriminative ability linearly increases with the increasing volume of training videos used (10M → 200M). Meanwhile, Figure 8 shows that the retrieval performance increase becomes marginal when scaling the training data beyond 50M. It's vital to note our model is trained using only contrastive loss without employing popular designs such as matching head and its corresponding loss. Consequently, this retrieval result doesn't allow for any definitive conclusions about whether there exists a turning point after which scaling up the training videos becomes less beneficial currently. More explorations are necessary in these retrieval experiments. However, these findings generally suggest that enhancing the scale of pretraining data can improve the transferability of the learned representation.

## 5.2 OTHER APPLICATIONS

**Text-to-Video Generation.** Our InternVid dataset improves existing text-to-video (t2v) generation models by providing high-quality video-text pairs. We extend spatiotemporal modeling on the latent space of a text-to-image diffusion model (Rombach et al., 2022) as a t2v baseline. We train the baseline with two settings: one using WebVid10M, and the other using InternVid-Aes-18M in addition to WebVid10M. InternVid-Aes-18M is a subset of InternVid consisting of clips with an aesthetic

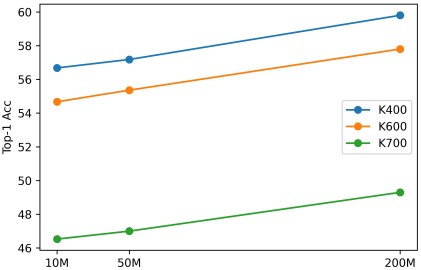 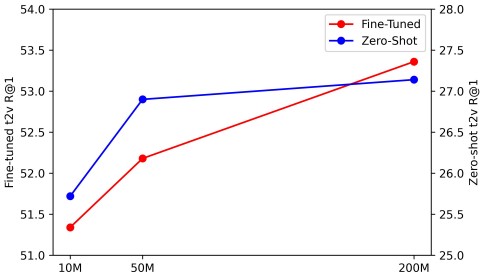

Figure 7: Zero-shot action recognition (top-1 accuracy) on Kinetics-400 / -600 / -700.

Figure 8: Video retrieval average performance (text-to-video R@1) across five datasets.

| Method | Training Data | IS ($\uparrow$) | UCF-101 FID ($\downarrow$) | FVD ($\downarrow$) | MSR-VTT CLIPSIM ($\uparrow$) |
|---|---|---|---|---|---|
| VideoCrafter[2] | WebVid10M (Bain et al., 2021) | 18.26 | 66.95 | 910.87 | 0.2875 |
| VideoFusion[3] | WebVid10M (Bain et al., 2021) | 17.49 | 75.77 | 639.90 | 0.2795 |
| t2v baseline | WebVid10M (Bain et al., 2021) | 13.97 | 98.25 | 705.25 | 0.2657 |
| t2v baseline | WebVid10M+InternVid18M | $21.04_{+7.07}$ | $60.25_{-38.00}$ | $616.51_{-88.74}$ | $0.2951_{+0.0294}$ |

Table 6: Zero-shot text-to-video generation performance.

score of at least 4. Quantitative (Table 6) and qualitative (Figure 12) evaluations demonstrate the effectiveness of InternVid in video generation tasks. Evaluation protocols are given in App. E.3.

In Table 6, we observe that t2v baseline trained on WebVid10M performs poorly in terms of IS, FID, and CLIPSIM when compared to other approaches. However, with the addition of InternVid-Aes, t2v baseline demonstrates significant improvements in these metrics and outperforms other methods by a considerable margin. In Figure 12, we observe that the t2v baseline using both WebVid10M and InternVid-Aes-18M significantly outperforms others in visual quality and temporal coherence. These results demonstrate the potential of InternVid for high-quality video generation.

**Video-Centric Dialogue System.** Inspired by recent vision-centric dialogue systems (Li et al., 2023c; Muhammad Maaz & Khan, 2023; Li et al., 2023a), we integrate our pretrained ViCLIP (with InternVid) into VideoChat (Li et al., 2023c) to show how our data and model can empower multimodal dialogue methods with effective video modeling capability. In implementation, we inherit nearly all designs of VideoChat-Embed, just replacing its visual encoder with our ViCLIP (trained on InternVid). We evaluate VideoChat-ViCLIP in spatial understanding (Figure 13), action recognition (Figure 14), temporal understanding (Figure 15), video reasoning (Figure 16), and video creative (Figure 17) tasks. Our qualitative evaluations demonstrate its decent video-to-text capabilities.

| Evaluation Aspect | Correctness of Information | Detail Orientation | Contextual Understanding | Temporal Understanding | Consistency | Avg |
|---|---|---|---|---|---|---|
| VideoChat (Eva-g) | 2.23 | 2.50 | 2.53 | 1.94 | 2.24 | 2.29 |
| Video-ChatGPT | 2.40 | 2.52 | 2.62 | 1.98 | 2.37 | 2.38 |
| VideoChat-ViCLIP | **2.86** | **2.52** | **3.08** | **2.36** | **2.40** | **2.64** |

Table 7: Performance benchmarking of text generation models.

In terms of quantitative comparison, as shown in Table 5.2, VideoChat-ViCLIP notably outperforms the vanilla VideoChat (using Eva-g as the vision encoder) and others across all evaluation aspects of the video conversation evaluation in the work of Muhammad Maaz & Khan (2023). Specifically, the model shows remarkable improvements in the correctness of information (from 2.23 to 2.86), contextual understanding (from 2.53 to 3.08), and temporal understanding (from 1.94 to 2.36). The average score also increases from 2.29 to 2.64, showing an overall performance gain.

# 6 CONCLUSION

Our dataset, InternVid, is designed for multimodal research (both understanding and generation) focused on videos. It consists of over 200 million video clips sourced from 7 million high-resolution YouTube videos. We use existing models with a multiscale approach to generate clip-level descriptions. Our studies confirm the efficacy of captions, and the large volume of video-text data enables crossmodal learning and text-to-video generation at scale. By training with our data, we develop a video-text representation baseline ViCLIP using ViT-L and analyze briefly how the data scale affects learned crossmodal embeddings. In addition to perception tasks, we show that InternVid improves text-to-video generation and supports multimodal dialogue systems. With its data, annotations, metadata, and computed scores, we believe InternVid can fuel a variety of studies and applications.

ACKNOWLEDGEMENTS

This work is partially supported by the National Key R&D Program of China (No. 2022ZD0160101), National Natural Science Foundation of China (No. 62076119, No. 61921006)), the Science and Technology Commission of Shanghai Municipality under Grant No. 23QD1400800 and No. 23YF1461900, and the Ministry of Education, Singapore, under its MOE AcRF Tier 2 (MOE-T2EP20221- 0012) and NTU NAP.

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

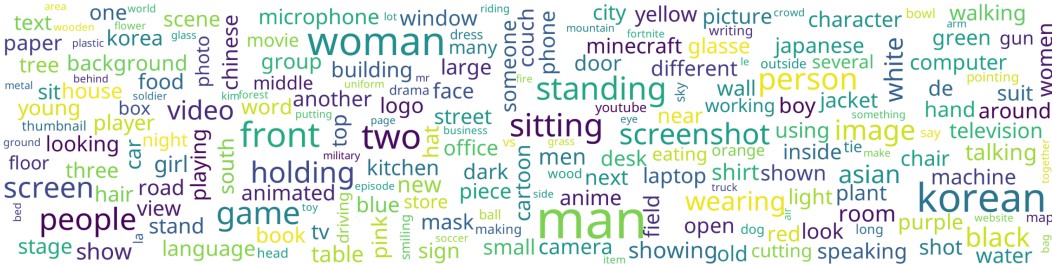

Figure 9: The word cloud (Top-200) of the generated captions in the InternVid dataset reveals that the captions predominantly highlight the rich actions of the objects.

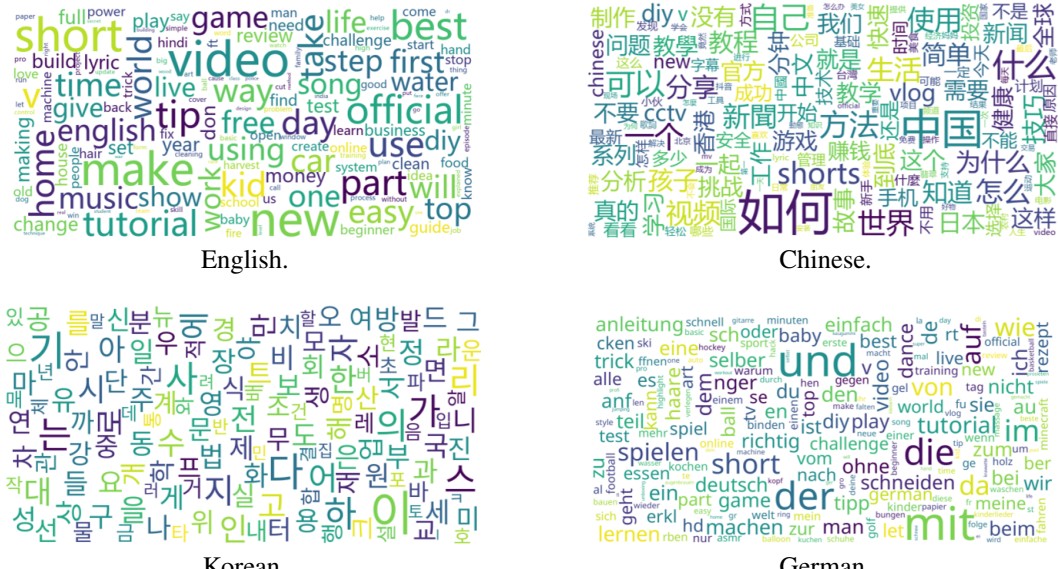

Figure 10: The word clouds of the ASR transcripts of four different languages (English, Chinese, Korean, and German). We collect videos from various countries or regions with 11 different languages. Here we list four of them to show how these transcripts are distributed in words.

## A  DATA AVAILABILITY STATEMENT

We are committed to maintaining transparency and compliance in our data collection and sharing methods. In accordance with these principles, please note the following:

**Publicly Available Data:** The data utilized in our studies is publicly available. We do not use any exclusive or private data sources.

**Data Sharing Policy:** Our data sharing policy builds upon the precedent set by prior works like Kinetics, HD-VILA, and others. Instead of providing the original raw data, we only supply the YouTube video IDs necessary for downloading the respective content.

**Usage Rights:** The data released by us is intended exclusively for research purposes. Any potential commercial usage is not sanctioned under this agreement.

**Compliance with YouTube Policies:** Our data collection and release practices are strictly in accord with YouTube's data privacy policies. We ensure that no user data or privacy rights are violated during the process.

**Data Licence:** We employ the protocol of CC BY 4.0.

## B    Limitations & Societal Impact

All video data used in our research are downloaded from YouTube using Safe for Work (SFW) queries and channels. To ensure appropriate content, we employ a simple NSFW filter: a binary classifier designed to recognize and exclude non-ethical videos. For privacy considerations and in respect of data sharing practices, we share only the YouTube ID of the videos, similar to previous academic works. This approach aligns with YouTube's data protocols and ensures no violation of privacy or data usage rules. Despite these precautions, our work has some limitations, primarily related to data diversity and representativeness. Although YouTube is an extensive source encompassing a wide range of video categories, certain specific types of footage may be excluded or scarcely collected, including: public area surveillance, sports competitions, movies, documentaries, etc. The exclusion of such categories is often due to copyright restrictions or other limits imposed by the platform. Therefore, while our dataset provides a broad view of everyday video content, its coverage does not extend to every possible category or type of video. These limitations should be taken into account when considering the generalizability of our results across all types of video data.

## C    More Statistics in InternVid

**Actionness.**    InternVid contains way more verbs than the WebVid10M. We used NLTK toolkit to analyze the number of verbs in captions, focusing on tagging all unique verbs. We found a total of 109,485 verbs in the WebVid10M, while InternVid contained 212,155 ones. While the counts may not be that accurate due to our simple counting, we believe they provide a rough indication of the actionness of the two datasets.

**Video Caption and Transcript Distribution.**    To analyze the word distribution of our generated captions and multilingual (ASR) transcripts, we compute their distributions. The resulting word distribution of the captions is presented in Figure 9, which includes objects (tv, car, door, plant, etc.), attributes (green, young, large, long, etc.), locations (middle, behind, south, next, etc.), scenes (room, stage, kitchen, office, etc.), actions/events (walking, eating, cutting, holding, etc.), and more.

We also include four word distributions of different languages in Figure 10, reflecting trends in different countries and offering potential data customization along with the provided metadata.

**Aesthetic Scores and Clip-Caption Similarity.**    We uniformly sampled four frames of each clip, calculated their aesthetic scores, and took the maximum score as the video aesthetic score. For clip-caption similarity, we computed the cosine similarity between video embeddings and text ones, again using a uniform sampling of four frames for each clip.

**Potential Biases.**    We focus on age, gender, and race distributions, as these are commonly recognized areas where bias can occur. Our methodology consisted of counting keywords related to these categories in the generated video captions. It's important to note that these synthetic captions may not fully reflect the truth of the corresponding videos, thereby creating a gap between our analysis and the actual reality.Here are the results of our analysis:

- Age distribution: We counted nouns related to children, grown-ups, and the elderly. We found that 30.71% of the video captions contained such descriptions. Within this subset, the majority were about adults (84.59%), followed by children (15.31%) and barely any mentions of senior citizens (0.08%).

- Gender distribution: We counted nouns specifically related to males and females. According to our findings, 33.7% of video captions contained some form of gender-related text. Among these, 64.27% pertained to men and 35.73% pertained to women.

- Race distribution: Only around 2.51% of video captions contained descriptions related to race. This could be due to the limitations of our captioning pipeline, which might not be capable of capturing such attributes accurately. Further exploration using a dedicated race recognition model is needed for more accurate statistics.

# D INTERNVID-ICL: INTERLEAVED VIDEO-TEXT FOR IN-CONTEXT VIDEO LEARNING

```
[..., "the inside of a home has a rug and a light on.", "♪
We could leave the Christmas lights up til January ♪", ...,
"woman with blond hair playing guitar", "♪ Have I known you

20 seconds or 20 years?  ♪",                        ,
"close-up of a bathroom sink with soap bubbles and other
items", "a bathroom is seen with a sink and two lights", "a
woman swiming inside of a fishbowl with a ladder and a man",

"♪ Can I go wher you go?  ♪",
, "devils roll the dice, angels roll their eyes","♪ And,
take me out, and take me home ♪" ,..., "the man is standing
in a room with pink carpet","♪ You're my, my ♪", "a woman
in yellow is dancing with a man in a red room", "♪ My, My
lover ♪",

                        , "a woman is sitting on a chair,
playing a guitar and a woman holding a balloon", "♪ ♪ ♪",
"two men smiling while holding wine glasses and drinking
beer", "♪ We could let our friends crash in the living room
♪" ...]
```

Table 8: **Interleaved video-text data format (b) in InternVid.** The caption and ASR transcript of each clip is shown in black and gray, respectively. We can achieve interleaved video-text data format (a) by abandoning ASR transcripts. To obtain data format (c), we concatenate multiple videos with interleaved video-text data (a).

**Visual Examples.** As given in the paper, we provide examples video+text interleaved entries for in-cntext learning as Flamingo. Table 8 gives an example about format (a): arrange clips and their descriptions sequentially based on their temporal order within the same video. Note the videos are randomly dropped with a probability (0.3) for constructing richer text context compared with the original video-text pair combinations in sequential.

# E IMPLEMENTATION DETAILS

## E.1 DIVERSITY SAMPLING IN CONSTRUCTING INTERNVID SUBSETS

For DIV (diversity sampling), we aim to sample video clips from all long videos available to maximize data diversity. This was done by counting the frequencies of long videos in the segmented clip pool and sampling clips with probabilities inverse to these frequencies. Here is a pseudocode example of this process:

```
1  from collections import Counter
2  import json
3  import random
4  import numpy as np
5  data = json.load(open("/path/to/to_sample"))
6  video_id = set([x["video"].split("/")[-1][:11] for x in data])
7  video_id_counter = Counter([x["video"].split("/")[-1][:11] for x in data
      ])
8  sampling_weights = [1.0 / video_id_counter[x["video"].split("/")
      [-1][:11]] for x in data]
9  np.random.seed(42)
10 sampling_weights = np.array(sampling_weights)
```

| config | MSRVTT | DiDeMo | ANet | LSMDC | MSVD |
|---|---|---|---|---|---|
| optimizer | | | AdamW | | |
| optimizer momentum | | | $\beta_1, \beta_2$=0.9, 0.999 | | |
| weight decay | | | 0.02 | | |
| learning rate schedule | | | cosine decay | | |
| learning rate | 2e-5 | 4e-5 | 2e-5 | 2e-5 | 4e-5 |
| batch size | | | 256 | | |
| warmup epochs | | | 1 | | |
| total epochs | 7 | 8 | 5 | 10 | 20 |
| input frame | | | 12 | | |
| max text length | 32 | 96 | 64 | 64 | 150 |
| drop path | 0.3 | 0.2 | 0.3 | 0.3 | 0.2 |
| flip augmentation | | | yes | | |
| augmentation | | MultiScaleCrop [0.5, 1] | | | |

Table 9: **Video-text retrieval fine-tuning settings.**

```
11  sampling_weights = sampling_weights / sampling_weights.sum()
12  sampled_index = np.random.choice(len(data), 10647458, replace=False, p=
        sampling_weights)
13  data = [data[i] for i in sampled_index]
14  json.dump(data, open("/path/to/sampled", "w"))
```

For FLT (filtering), we applied a series of filtering strategies to video data alongside DIV sampling. These included: a) Removing video clips shorter than 1s (approximately 23.15% of the total) or longer than 120s (around 0.84% of the total). b) Computing CLIPScore for each video clip using a randomly sampled frame from the clip with OpenAI's CLIP-ViT-L/14, then selecting clips within the top 30% of CLIPScores. c) Sampling 10M out of the remaining clips using DIV sampling.

### E.2 VICLIP

**Implementation.** ViCLIP is learned with 64 NVIDIA A100 GPUs for 3 days with 50M video-text pairs. We introduce DeepSpeed and FlashAttention (Dao et al., 2022) for training and inference.

**Action Recognition.** In the zero-shot action recognition, we sample 8 frames in each video. Following the settings in CLIP and EVA-CLIP, we report the mean of top-1 and top-5 accuracy for Kinetics-400 / -600 / -700. In Section 5.1, we show ViCLIP learnt on WebVid or InternVid is an effective zero-shot action recognition model.

In the full fine-tuned setting, we conduct two experiments with two receipts. In Table 3, for the experiments where the training data excluded K710, we followed the common practice of finetuning the pretrained ViCLIP with the training data from the evaluation dataset. On the other hand, for the experiments where the training data included K710, we adopted a training trick inspired by (Li et al., 2022a). We first finetuned the pretrained ViCLIP with K710 (Li et al., 2022a), and then proceeded with the common supervised finetuning setting. By incorporating the supervised finetuning with K710, ViCLIP demonstrated better performance in the fine-tuned tasks compared to experiments that did not include K710.

**Video Retrieval.** In the full-finetuning setting, we tune the pretrained ViCLIP with not only video-text contrastive loss but also video-text matching loss on the training data of the evaluated benchmarks. During both training and testing, we sample 12 frames. Detailed hyper-parameters are given in Table 9. In the zero-shot setting, we sample only 8 frames for evaluations.

### E.3 VIDEO GENERATION BASELINE

We used the spatiotemporal modeling approach from (Wu et al., 2022) and built our text-to-video generation baseline on the work of (Rombach et al., 2022). Our approach consists of a U-Net with a transformer that models its latents, using interleaved spatiotemporal attention (ST-Attn), cross-attention for visual-text, a feed-forward network (FFN), and temporal attention (T-Attn), as illustrated in Figure 11. To adapt the 2D convolutional layers in (Rombach et al., 2022) to 3D, we extended $3 \times 3$ kernels into $1 \times 3 \times 3$ ones. We also extended the original spatial attentions to spatiotemporal ones. We initialized our baseline using all text-to-image diffusion model parameters, while the newly added temporal attention layers used default parameters.

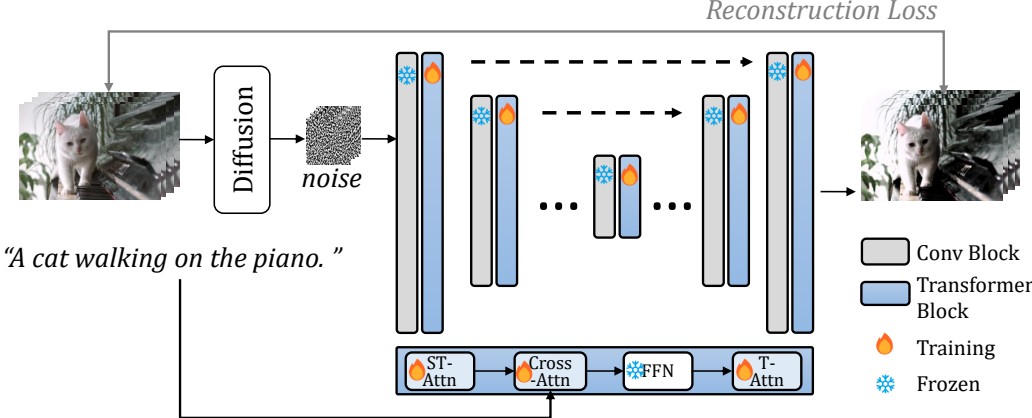

Figure 11: Framework of our text-to-video generation baseline.

| Captioning | Retrieval | | | | Action Recognition | | | | | |
|---|---|---|---|---|---|---|---|---|---|---|
| | Zero-Shot | | Fine-Tuned | | Zero-Shot | | | | | |
| Method | MSR-VTT | | MSR-VTT | | K400 | | K600 | | K700 | |
| | T2V | V2T | T2V | V2T | top-1 | AVG | top-1 | AVG | top-1 | AVG |
| VideoChat | 33.9 | 32.3 | 46.6 | 47.1 | 54.68 | 67.74 | 51.70 | 64.91 | 43.67 | 56.51 |
| Ours | **38.6** | **38.5** | **49.0** | **49.2** | **58.52** | **71.11** | **55.37** | **68.27** | **47.09** | **59.98** |

Table 10: Video retrieval and action recognition results of ViCLIP-B trained on InternVid-FLT-10M with the captions generated by VideoChat and our captioning approach.

For the ST-Attn implementation, we used frame embeddings from the U-Net encoder instead of video embeddings as in (Wu et al., 2022). We concatenated the embeddings of the previous and current frame for values and keys in attention, while using the current frame embedding alone as queries. The rest of the implementation remained the same as the original.

**Text-to-Video Evaluation.**    To evaluate our text-to-video model, we conducted zero-shot experiments on the UCF-101 and MSRVTT datasets, following the method from (Blattmann et al., 2023). For UCF-101, we used the class names as text prompts and generated 20 samples per class (total of 2,020 videos). For MSRVTT, we randomly selected one caption per video from the official test set (total of 2,990 videos). To ensure a fair comparison, we used the official implementation of VideoCrafter and VideoFusion (Luo et al., 2023) to generate the same number of videos with the same text prompts. During video sampling and evaluation, we generated 16 frames per video.

We assess the overall quality of the synthesized results on UCF-101 using framewise-FID, FVD, and Inception Score (IS), and evaluate the text-video semantic similarity on MSRVTT using clip similarity (CLIPSIM). For framewise-FID and IS, we use the pretrained Inceptionv3 network weights as our image encoder. For FVD, we use the pretrained InceptionI3d model and followed the TATS method (Ge et al., 2022). To compute CLIPSIM, we calculate the clip text-image similarity for each frame with respect to the given text prompts and computed the average score. We use the ViT-B-32 clip model as the backbone, consistent with previous work (Blattmann et al., 2023).

## F    MORE RESULTS

### F.1    EFFECTIVENESS OF OUR MULTISCALE CAPTIONING APPROACH

To further validate the effectiveness of our proposed captioning method, we establish a video caption baseline using the video multimodal model VideoChat (Li et al., 2023c) for comparison. We input the video clip into the model with the prompt `"Please describe the content in the given video."` and apply it to InternVid-10M-FLT, resulting in 10 million new captions generated by VideoChat. Subsequently, we train two versions of ViCLIP-Base using InternVid-10M-FLT, each version trained with one of the two types of captions.

Table 10 demonstrates that ViCLIP-B trained using our captions outperforms the version trained using captions from VideoChat in both video retrieval (MSR-VTT) and action recognition (K400/600/700). These results are particularly noteworthy considering that the only difference in training lies in the captions generated by the

| Method | Data | K400 | | K600 | | K700 | |
| --- | --- | --- | --- | --- | --- | --- | --- |
| | | top-1 | AVG | top-1 | AVG | top-1 | AVG |
| ViCLIP | +InternVid-2M | 51.70 | 64.69 | 49.20 | 62.34 | 40.90 | 53.70 |
| ViCLIP | +InternVid-2M-BLIP2 | 38.40 | 51.58 | 36.40 | 49.19 | 29.10 | 40.68 |

Table 11: Zero-shot action recognition results of ViCLIP using different captions on Kinetics 400/600/700.

| Method | Data | MSR-VTT | | LSMDC | | DiDeMo | | MSVD | | ANet | |
| --- | --- | --- | --- | --- | --- | --- | --- | --- | --- | --- | --- |
| | | T2V | V2T | T2V | V2T | T2V | V2T | T2V | V2T | T2V | V2T |
| ViCLIP | +InternVid-2M | 31.8 | 33.7 | 14.3 | 12.7 | 13.6 | 21.5 | 39.6 | 62.5 | 9.9 | 16.8 |
| ViCLIP | +InternVid-2M-BLIP2 | 21.7 | 21.9 | 5.2 | 5.1 | 7.1 | 12.7 | 24.6 | 42.1 | 6.4 | 10.2 |

Table 12: Results of zero-shot video retrieval from ViCLIPusing different captions on MSR-VTT, LSMDC, DiDeMo, MSVD, and ActivityNet.

two different approaches. Therefore, these findings further confirm the superior performance of our proposed captioning method compared to the baseline VideoChat.

We also ablate the necessary of including fine-level captions (by tag2text) more than only using coarse ones (by BLIP2). Specifically, an ablation is performed on two subsets of the dataset (InternVid-2M and InternVid-2M-BLIP), each having 2 million video-text pairs. InternVid-2M utilized fused captions, combining both coarse- and fine-level ones. In contrast, InternVid-2M-BLIP only used the coarse-level captions produced by BLIP2 on the central frames. For the mentioned using fine-level captions, concatenating the framewise captions from tag2text as the video captions is not a promising opinion as these captions are quite long and full of reptitions, unsuitable for contrastive learning. Thus, we do not include this setting in experiments.

Zero-shot experiments are conducted on these models. Due to computational constraints, the ViCLIP-B is trained with a batch size of 4096 using 8 A100 GPUs, with a mask ratio set to 0.9. All remaining training parameters were consistent with those in the main paper. Contrasting the results from Table 11 and 12, it's evident that the use of combined coarse and fine-level captions in video-text contrastive learning rendered superior zero-shot performance than utilizing the coarse level ones alone. It shows the effectiveness of our given video captioning pipeline.

**How the LM impacts motion-related words when summarizing framewise captions into video ones.** From a statistical perspective, generating video captions from frame-level captions using a language model has a negligible effect on the number of motion-related words captured for video-based understanding. To illustrate this, we counted the unique verbs (using nltk package) in the captions from a 10m subset of InternVid under two settings: 1) In the first setting, the captions are video captions generated by the language model. 2) In the second setting, the captions are frame-wise ones from BLIP2 and tag2text. We found that the number of unique verbs in the video captions is 109859, whereas for the frame-wise captions it is slightly higher at 109895. This small discrepancy suggests that almost no motion-related words are lost during the caption generation process by LM. Therefore, we believe our approach maintains most of the important motion-related information needed for video understanding.

## F.2 MODEL SCALING

We provide a comparison between two versions of ViCLIP, -L (300M) and -B (80M), when trained on InternVid in Tables 13, 14, and 15. These tables distinctly demonstrate that moving from the base to large model, ViCLIP's zero-shot and finetuned video retrieval performance, as well as zero-shot action recognition, can be consistently improved. These tables clearly demonstrate the benefits of model scaling, and we aim to explore this area further in future work as resources permit.

## F.3 LINEAR PROBING

we present the linear action recognition results on Kinetics-400 in Table 16. It's noteworthy that ViCLIP, trained on InternVid-10M-FLT/-200M, delivers a much higher top-1 accuracy compared to when trained on WebVid-10M (with a more than 10-point increase), mirroring our findings in fine-tuned action recognition settings. Comparing with other approaches, ViCLIP-L offers performance close to TVTSv2-H/-B, which incorporate extra learnable parameters for spatiotemporal modeling. Moreover, it significantly outperforms VideoMAEv2-H. This result can be attributed to the fact that MAE-based methodologies generally underperform in linear evaluations.

| Method | Data | MSR-VTT | | LSMDC | | DiDeMo | | MSVD | | ANet | |
|---|---|---|---|---|---|---|---|---|---|---|---|
| | | T2V | V2T | T2V | V2T | T2V | V2T | T2V | V2T | T2V | V2T |
| ViCLIP-B | +InternVid-200M | 37.4 | 36.1 | 16.5 | 15.2 | 16.6 | 22.6 | 44.3 | 67.0 | 13.3 | 21.7 |
| ViCLIP-L | +InternVid-200M | 39.3 | 39.5 | 18.3 | 16.6 | 17.1 | 25.5 | 47.3 | 70.0 | 13.7 | 21.6 |
| ViCLIP-B | +InternVid-10M-FLT | 38.6 | 38.5 | 18.5 | 17.0 | 16.3 | 25.0 | 44.8 | 67.2 | 13.0 | 21.8 |
| ViCLIP-L | +InternVid-10M-FLT | 42.4 | 41.3 | 20.1 | 16.9 | 18.4 | 27.9 | 49.1 | 75.1 | 15.1 | 24.0 |

Table 13: Scaling model in zero-shot video retrieval on MSR-VTT, LSMDC, DiDeMo, MSVD, and ActivityNet.

| Method | Data | MSR-VTT | | LSMDC | | DiDeMo | | MSVD | | ANet | |
|---|---|---|---|---|---|---|---|---|---|---|---|
| | | T2V | V2T | T2V | V2T | T2V | V2T | T2V | V2T | T2V | V2T |
| ViCLIP-B | +InternVid-200M | 50.7 | 49.4 | 25.3 | 25.4 | 41.1 | 40.8 | 69.0 | 69.3 | 37.7 | 35.8 |
| ViCLIP-L | +InternVid-200M | 53.7 | 53.4 | 29.3 | 29.3 | 51.1 | 50.8 | 79.9 | 78.4 | 52.8 | 51.1 |
| ViCLIP-B | +InternVid-10M-FLT | 49.0 | 49.2 | 24.4 | 23.7 | 40.0 | 41.4 | 72.2 | 73.7 | 38.3 | 37.0 |
| ViCLIP-L | +InternVid-10M-FLT | 52.5 | 51.8 | 33.0 | 33.0 | 49.4 | 50.2 | 77.2 | 79.0 | 49.8 | 48.1 |

Table 14: Scaling model in fine-tuned video retrieval on MSR-VTT, LSMDC, DiDeMo, MSVD, and ActivityNet.

## F.4 Impact of Videos from Different Language Sources

Unlike previous models that are benchmarked mostly on English-based datasets, InternVid encompasses clips from a variety of languages. This necessitates a further analysis to determine the potential impact this diversity might have. Currently, we hypothesize that the language of the video may not significantly impact the generated captions as our deployed image caption models generate English descriptions based purely on input frames. However, in terms of video distributions, there may exist differences (such as in behaviors, activities, and events) between videos stemming from different countries due to varied cultural backgrounds.

To examine this hypothesis, we select two 2 million subsets of InternVid: one consisting of only English videos (InternVid-2M-EN) and another with only Chinese videos (InternVid-2M-CN). It's important to note that whether the videos are in English or Chinese, we generate captions in English. Our ViCLIP-B model is pretrained on these subsets, and we conduct zero-shot experiments as described below. Due to resource constraints, we trained the ViCLIP-B with a batchsize of 4096 using 8 A100 GPUs with a mask ratio set to 0.9. The remaining training settings were kept consistent with those outlined in the paper. As given in Table 17 and 18, we find that the model pretrained with InternVid-2M-EN outperformed that with InternVid-2M-CN notably in both zero-shot action recognition on K400/600/700 and video retrieval. This result can be attributed to the fact that InternVid-2M-EN has a data distribution much closer to downstream task data than InternVid-2M-CN, as all used task videos are sourced from English sources.

## F.5 Text-to-Video Generation

In Figure 12, we observe that the t2v baseline using both WebVid10M and InternVid-Aes-18M significantly outperforms others in visual quality and temporal coherence. Note that the t2v baseline using InternVid does not contain watermarks, which is a data bias in WebVid10M. We give more visual comparisons between our baseline and other approaches in the supp.

## F.6 Video-Centric Dialogue System

We give qualitative evaluations of VideoChat-ViCLIP in spatial understanding (Figure 13), action recognition (Figure 14), temporal understanding (Figure 15), video reasoning (Figure 16), and video creative (Figure 17) tasks.

| Method | Data | K400 top-1 | K400 AVG | K600 top-1 | K600 AVG | K700 top-1 | K700 AVG |
|--------|------|------------|----------|------------|----------|------------|----------|
| ViCLIP-B | +InternVid-200M | 56.58 | 69.20 | 53.57 | 66.20 | 45.82 | 58.28 |
| ViCLIP-L | +InternVid-200M | 59.80 | 71.09 | 57.80 | 69.34 | 49.30 | 61.25 |
| ViCLIP-B | +InternVid-10M-FLT | 58.52 | 71.11 | 55.37 | 68.27 | 47.09 | 59.98 |
| ViCLIP-L | +InternVid-10M-FLT | 64.80 | 75.70 | 62.20 | 73.53 | 54.30 | 66.38 |

Table 15: Scaling model in zero-shot action recognition results on Kinetics 400/600/700.

| Method | Data | K400 top-1 | K400 AVG |
|--------|------|------------|----------|
| VideoMAE-B (Tong et al., 2022) | Kinetics-400 | 20.4 | - |
| VideoMAEv2-H (Wang et al., 2023) | Kinetics+SthSth+AVA+WebVid2M | 25.8 | - |
| TVTS-B (Zeng et al., 2023b) | +YT-Temporal-180M | 60.8 | - |
| TVTSv2-B (Zeng et al., 2023a) | +YT-Temporal-180M+WebVid-2M | 70.1 | - |
| TVTSv2-H (Zeng et al., 2023a) | +YT-Temporal-180M+WebVid-2M | 73.1 | - |
| ViCLIP-L | +WebVid-10M | 60.0 | 82.9 |
| ViCLIP-L | +InternVid-10M-FLT | 71.1 | 90.4 |
| ViCLIP-L | +InternVid-200M | 71.7 | 90.9 |

Table 16: Linear action recognition results on Kinetics-400.

| Method | Data | K400 top-1 | K400 AVG | K600 top-1 | K600 AVG | K700 top-1 | K700 AVG |
|--------|------|------------|----------|------------|----------|------------|----------|
| ViCLIP | +InternVid-2M-EN | 40.20 | 53.68 | 37.40 | 51.00 | 29.60 | 42.06 |
| ViCLIP | +InternVid-2M-CN | 35.9 | 49.73 | 33.70 | 47.05 | 26.90 | 39.02 |

Table 17: Zero-shot action recognition results of ViCLIP using different pretraining sources on Kinetics 400/600/700.

| Method | Data | MSR-VTT T2V | MSR-VTT V2T | LSMDC T2V | LSMDC V2T | DiDeMo T2V | DiDeMo V2T | MSVD T2V | MSVD V2T | ANet T2V | ANet V2T |
|--------|------|-------------|-------------|-----------|-----------|------------|------------|----------|----------|----------|----------|
| ViCLIP | +InternVid-2M-EN | 24.1 | 24.1 | 9.8 | 9.1 | 10.3 | 15.6 | 31.4 | 50.5 | 7.1 | 12.0 |
| ViCLIP | +InternVid-2M-CN | 22.2 | 22.2 | 8.4 | 8.7 | 9.9 | 15.1 | 29.6 | 48.5 | 5.7 | 9.7 |

Table 18: Results of zero-shot video retrieval of ViCLIP using different pretraining sources on MSR-VTT, LSMDC, DiDeMo, MSVD, and ActivityNet.

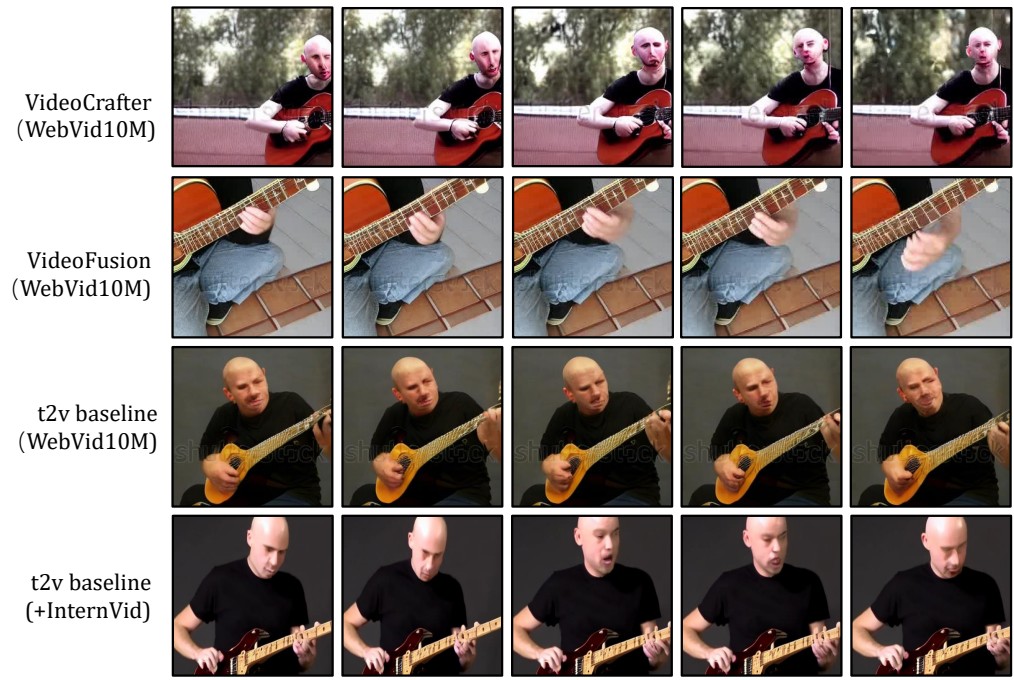

Figure 12: Comparison of samples from t2v baseline to others. We provide zero-shot text-to-video generation results of different methods trained on both WebVid10M and the additional InternVid-Aes-18M. The used prompt is: `a bald man in a black t-shirt is playing a guitar.`

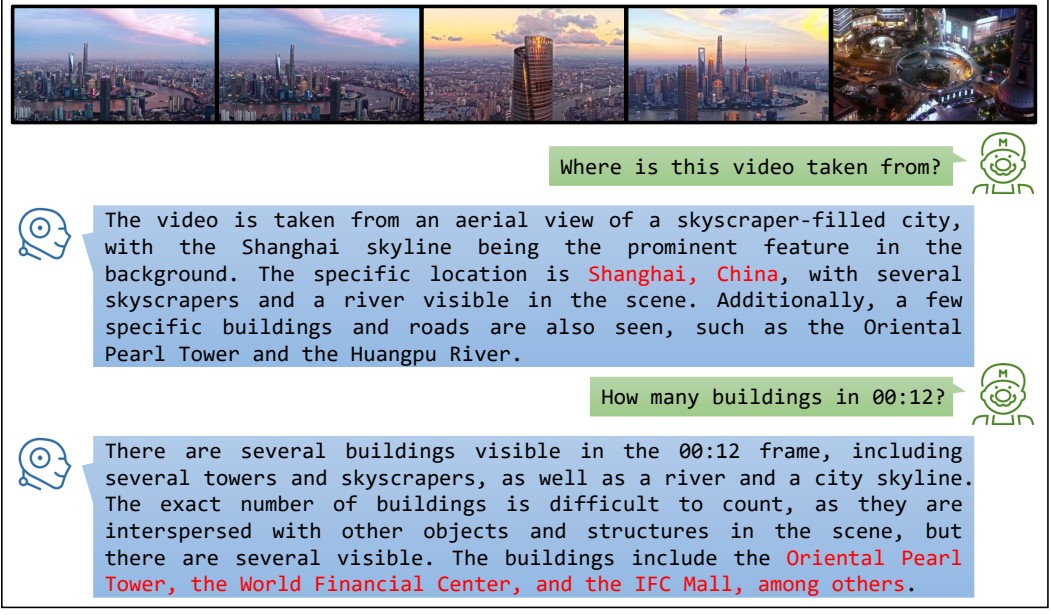

Figure 13: **Video spatial understanding task.** This figure demonstrates the spatial understanding and temporal localization capabilities of VideoChat-ViCLIP trained with our dataset.

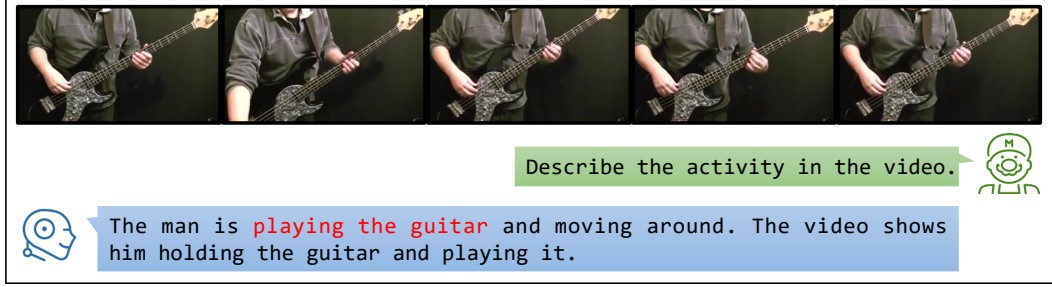

Figure 14: **Video action recognition task.** The video comes from Kinetics-400, with the label "playing guitar".

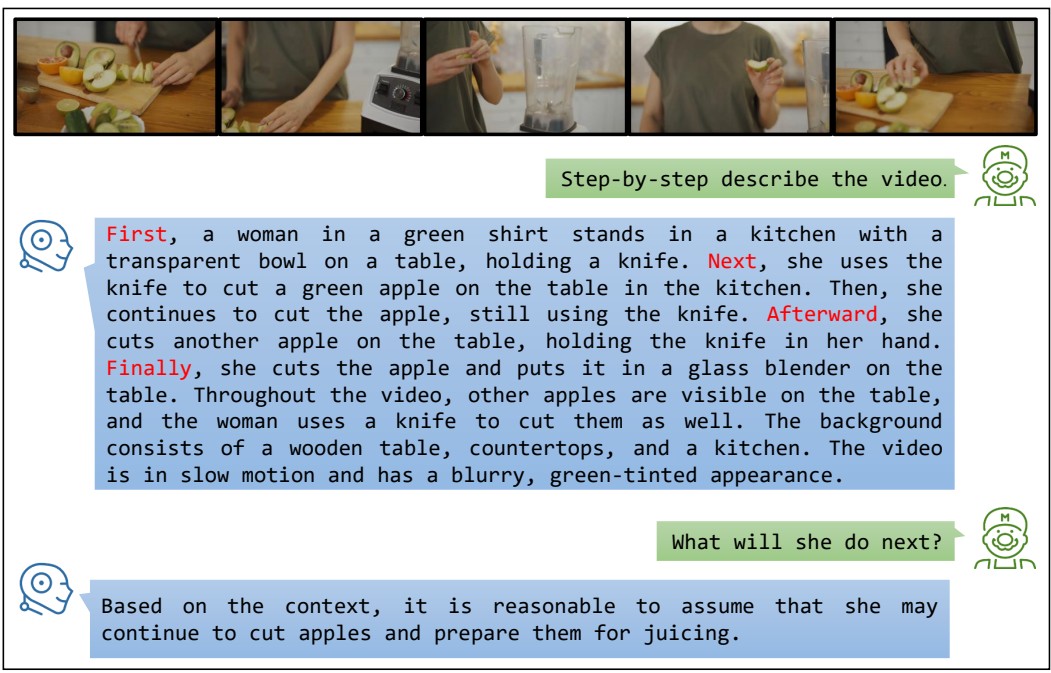

Figure 15: **Temporal understanding task.** VideoChat-ViCLIP can easily handle temporal understanding tasks and make predictions based on these temporal sequences due to its training on rich temporal information on InternVid.

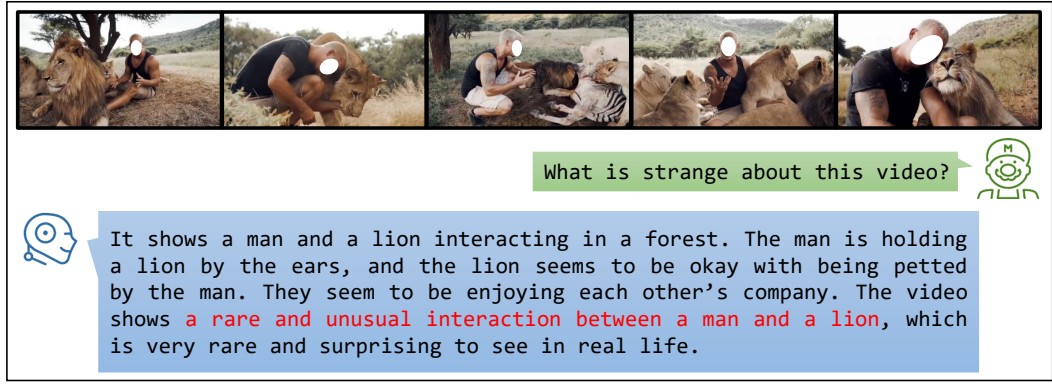

Figure 16: **Video reasoning task.** Our dialogue system has the ability to understand common sense and make causal inferences based on the contents of the video.

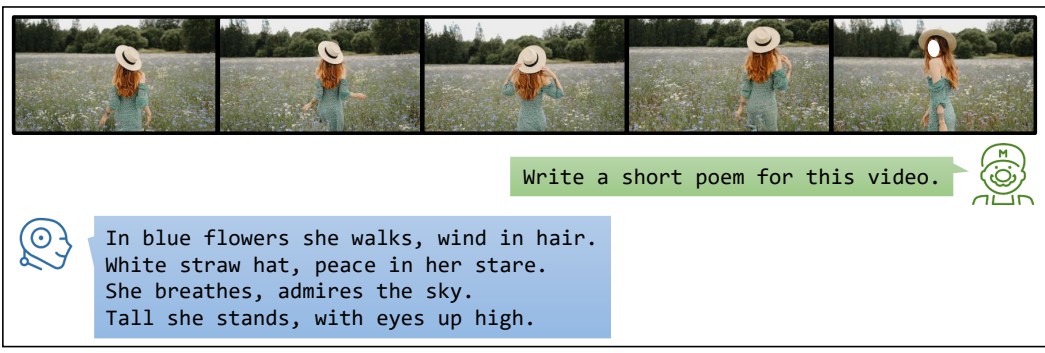

Figure 17: **Video creative task.** This figure shows the performance of VideoChat-ViCLIP on creative and generative tasks within our dataset, like generating poetry based on video content.

