# OpenReview forum: "InternVid: A Large-scale Video-Text Dataset for Multimodal Understanding and Generation"
_ICLR.cc/2024/Conference — ICLR 2024 spotlight_

### Official Review · Reviewer_1UqW · 2023-10-21

**Soundness:** 2 fair
**Presentation:** 2 fair
**Contribution:** 3 good
**Rating:** 6
**Confidence:** 4

**Summary:**

This paper presents InternVid, an large-scaled video-language dataset, consisting of a collection of 7M videos, 234M video clips and corresponding textual descriptions. They consider diversity and data quality by sourcing YouTube contents from various countries and languages. This new dataset enables pretraining a video-language model, named ViCLIP, exhibiting state-of-the-art scores in action recognition and video retrieval tasks. It also demonstrates a credible generative capability in text-to-video generation, supported by qualitative and quantitative assessment.

**Strengths:**

1. Clarity in writing

- The paper is well-written and easy to understand.

2. Scale and impact

- InternVid includes large-scaled 7.1M videos, making it a significant contribution to the video-language pretraining. Referring to table 1, InternVID comprises 234M 720p video clips with LLM generated captions which shows outperforming accuracy in description. The dataset’s substantial size can potentially have a profound impact on the field.

3. Diversity and Quality

- The effort to collect videos from various countries and languages is commendable. This diversity in data collection may address the issue of bias, a crucial aspect of large-scale datasets, as it mitigates the risk of training biased models.

4. High performance

- The suggested model, referred as ViCLIP, achieve the state-of-the-art in video-language benchmarks such as  action recognition and text-to-video retrieval.
- InternVID improves text-to-video generation by empowering the previous models with its large scale video-text dataset. It shows quantitative and qualitative (Figure 12) improvement in generation quality.

**Weaknesses:**

1. Lack of definitions, explanations

- Definition of UMT-SIM is absent. In table 2, justification of the design choice of InternVid-10M-FLT is not explained. (At glance, the result of performance improvement after filtering the dataset with a high UMT-SIM score seems trivial.)
- Figure 5 introduced 3 schemes of interleaving video clips, text, and ASR text. However, comparison between the methods or further analysis is not given.
- It is doubtful that CLIP is a suitable choice for comparing their method in action recognition tasks. There are likely more appropriate video models to use in the benchmark for a fairer comparison.

2. Lack of novelty

- The proposed method for generating the dataset is not novel. It simply uses BLIP to generate captions for video frames, and exploits LLM to generate a summarized caption for the video. Also, description of the used LLM is not given.
- ViCLIP is not a novel architecture and inherits the CLIP model without significant modifications. The paper predominantly focuses on introducing the dataset, which leaves room for exploring further architectural improvement and training techniques.

3. Unclear model selection process

- The paper lacks information on the criteria used to select the image captioning and language models.

4. Absence of ablation study and potential impact

- In contrast to previous models which are benchmarked in datasets mostly composed of English, InternVID consists of clips with diverse languages. Further analysis about the potential impact caused by this aspect should be considered.

5. Minor comments on Figure 2

- Similar colors like green and dark green are used, but this may cause confusion. Using more distinct and discrete colors, such as red and blue, could enhance the clarity.

**Questions:**

1. Consideration with Table 6

- Comment of VideoCrafter, VideoFusion is absent. Does InternVID also improve VideoCrafter, VideoFusion in IS, FID, FVD, and CLIPSIM? Can you provide further analysis about the choice of InternVID-Aes-18M rather than exploiting the full InternVID-200M?

2. Consideration with Section 5.2 and Figure 13~17

- To insist that InternVID provides more powerful video-text aligned representations for video-centric dialogue systems, authors must compare the results from VideoChat-ViCLIP with that of vanilla VideoChat. Can you provide quantitative and qualitative comparisons, showing improvement of the results after replacement of the previous visual encoder with ViCLIP?

**Details Of Ethics Concerns:**

The data source is YouTube, where the content often includes personal recordings, potentially raising privacy concerns. The collectors should carefully address these issues and rigorous data selection criteria should be put in place to ensure strict adherence to privacy and other relevant guidelines.

---

> ### Author Response · Authors · 2023-11-20
> **Response to Official Review of Submission4917 by Reviewer 1UqW: Q1**
>
> **Q1: Lack of definitions, explanations. Definition of UMT-SIM is absent. In table 2, justification of the design choice of InternVid-10M-FLT is not explained. (At glance, the result of performance improvement after filtering the dataset with a high UMT-SIM score seems trivial.) Figure 5 introduced 3 schemes of interleaving video clips, text, and ASR text. However, comparison between the methods or further analysis is not given. It is doubtful that CLIP is a suitable choice for comparing their method in action recognition tasks. There are likely more appropriate video models to use in the benchmark for a fairer comparison.**
>
> 1. UMT-SIM: In our work, UMT-SIM refers to the use of Unmasked Teacher (UMT) [R1] to compute the similarity score between a given video clip and the accompanying text. UMT is a video-language model that performs similarly to CLIP in terms of video-text similarity computation, but is trained in different data and task settings. Specifically, we extract the normalized video embedding of the given clip using UMT's video encoder and the normalized text embedding of the text using UMT's text encoder, then we compute their cosine similarity.
>
> 2. For InternVid-10M-FLT, we apply filtering strategies to video data in addition to DIV sampling. The employed steps as given as:
> a) Removing video clips shorter than 1s (approximately 23.15% of the total) or longer than 120s (around 0.84% of the total).
> b) Computing CLIPScore for each video clip using a randomly sampled frame from the clip with OpenAI's CLIP-ViT-L/14, then selecting clips within the top 30% of CLIPScores.
> c) Sampling 10M out of the remaining clips using DIV sampling.
>   For DIV (diversity sampling), we aimed to sample video clips from all long videos available to maximize data diversity. This was done by counting the frequencies of long videos in the segmented clip pool and sampling clips with probabilities inverse to these frequencies. Here is a pseudocode example of this process:
>
> ```python
>     from collections import Counter
>     import json
>     import random
>     import numpy as np
>
>     data = json.load(open("/path/to/to_sample"))
>     video_id = set([x["video"].split("/")[-1][:11] for x in data])
>     video_id_counter = Counter([x["video"].split("/")[-1][:11] for x in data])
>     sampling_weights = [1.0 / video_id_counter[x["video"].split("/")[-1][:11]] for x in data]
>     np.random.seed(42)
>     sampling_weights = np.array(sampling_weights)
>     sampling_weights = sampling_weights / sampling_weights.sum()
>     sampled_index = np.random.choice(len(data), 10647458, replace=False, p=sampling_weights)
>     data = [data[i] for i in sampled_index]
>     json.dump(data, open("/path/to/sampled", "w"))
> ```
>
> 3. Interleaving Video Clip Schemes: We introduced three potential schemes without claiming them as our contributions. The intention was to illustrate possible applications of the dataset.
>
> 4. Benchmark for Action Recognition Tasks: We acknowledge that CLIP might not be the most suitable model for action recognition. Therefore, we also compared our method with other specialized video models for action recognition (see Table A below). Notably, ViCLIP trained only using contrastive loss and without additional learnable parameters on InternVid outperforms both CLIP-ViP and TVTSv2, where the latter two approaches are also pretrained on large-scale video datasets, demonstrating the effectiveness of our dataset.
>
>   ### Table A. Zero-shot action recognition results on Kinetics 400/600/700.
>   |Method | Training Data | K400 |  | K600 |  | K700 |  |
>   |:---:|:---:|:---:|:---:|:---:|:---:|:---:|:---:|
>   | | |top-1 | AVG | top-1 | AVG | top-1 | AVG |
>   |CLIP-ViP-B [R2] | +HD-VILA-100M(sub+cap)+Img-text data | 37.6 | - | - | - | - | - |
>   |TVTSv2-B [R3] | +YT-Temporal-180M+WebVid-2M | 54.3 | - | - | - | - | - |
>   |ViCLIP-B | +InternVid-200M | 56.58 | 69.20 | 53.57 | 66.20 | 45.82 | 58.28 |
>   |ViCLIP-B | +InternVid-10M-FLT | 58.52 | 71.11 | 55.37 | 68.27 | 47.09 | 59.98 |
>   |ViCLIP-L | +InternVid-200M | 59.80 | 71.09 | 57.80 | 69.34 | 49.30 | 61.25 |
>   |ViCLIP-L | +InternVid-10M-FLT | 64.80 | 75.70 | 62.20 | 73.53 | 54.30 | 66.38 |
>   |TVTSv2-H [R3] | +YT-Temporal-180M+WebVid-2M | 59.6 | - | - | - | - | - |
>
> References:
>
> [R1] Unmasked teacher: Towards training-efficient video foundation models. In ICCV. 2023.
>
> [R2] CLIP-ViP: Adapting Pre-trained Image-Text Model to Video-Language Representation Alignment. In ICLR. 2023.
>
> [R3] TVTSv2: Learning Out-of-the-box Spatiotemporal Visual Representations at Scale. In arXiv. 2023.

---

> > ### Author Response · Authors · 2023-11-20
> > **Response to Official Review of Submission4917 by Reviewer 1UqW: Q2 & Q3**
> >
> > **Q2: Lack of novelty: The proposed method for generating the dataset is not novel. It simply uses BLIP to generate captions for video frames, and exploits LLM to generate a summarized caption for the video. Also, description of the used LLM is not given. ViCLIP is not a novel architecture and inherits the CLIP model without significant modifications. The paper predominantly focuses on introducing the dataset, which leaves room for exploring further architectural improvement and training techniques.**
> >
> > We appreciate your feedback and understand the concerns raised about novelty. However, it's important to highlight that our primary contribution centers around creating a scalable approach for constructing an efficient video-text dataset using Language Models (LM). This addresses a well-known data bottleneck in video-text pretraining.
> >
> > In comparison to image-text pretraining, video-text pretraining is less explored due to the dearth of high-quality, large-scale video-language datasets. Our approach addresses this gap by providing a means to generate such datasets efficiently and at scale.
> >
> > Our method utilizes both BLIP2 and tag2text for frame captioning to ensure effective video description generation—BLIP2 captions the central frame while tag2text performs framewise captioning.
> >
> > While we agree that ViCLIP is not entirely novel as it builds upon the CLIP model, it is essential to note that it serves as a demonstration of how our newly generated dataset can be effectively utilized. The experimental results support the efficacy of our method. ViCLIP, when pretrained on InternVid, has shown to achieve state-of-the-art zero-shot action recognition performance on Kinetics 400/600/700. Furthermore, ViCLIP pretrained on InternVid-10M-FLT/DIV significantly outperforms the same model trained on WebVid10M which uses alt-text as captions, indicating the superior quality of our dataset.
> >
> > As for further architectural improvements and training techniques, we absolutely agree that there is room for exploration. In this work, however, our focus is primarily on introducing and demonstrating the effectiveness of the dataset we have produced. Going forward, these enhancements would indeed constitute interesting avenues for future research.
> >
> > **Q3: Unclear model selection process: The paper lacks information on the criteria used to select the image captioning and language models.**
> >
> > BLIP2 and tag2text: These were chosen based on their exceptional performance in captioning tasks as validated on NoCaps and COCO datasets. The robustness of these models in generating accurate and meaningful captions was a key criterion in the selection process.
> >
> > T5-summary model: This model was picked after we found that the summary capabilities of open-sourced Language Model (LLM), Llama 1, and its variants did not meet our requirements. Specifically, during our work on InternVid starting in April, we performed qualitative evaluations and found that the T5 model outperformed existing LLMs such as Llama2-7b or Llama2-13b. The selection criteria were:
> >
> > 1. Speed: The T5 model showcased significantly faster processing speed compared to the Llama models. On average, T5 completed each summary within 14.96ms, while Llama2-7b took longer at 81.51ms. Both models were tested using an A100-80G.
> >
> > 2. Absence of hallucinations: Unlike existing LLMs, the T5 model did not incur any hallucinations during its operation, thus enhancing its reliability.
> >
> > We hope this clarifies the model selection process utilised in our research.

---

> ### Author Response · Authors · 2023-11-20
> **Response to Official Review of Submission4917 by Reviewer 1UqW: Q4 - Q6**
>
> **Q4: Absence of ablation study and potential impact. In contrast to previous models which are benchmarked in datasets mostly composed of English, InternVID consists of clips with diverse languages. Further analysis about the potential impact caused by this aspect should be considered.**
>
> Unlike previous models that are benchmarked mostly on English-based datasets, InternVID encompasses clips from a variety of languages. This necessitates a further analysis to determine the potential impact this diversity might have.
> Currently, we hypothesize that the language of the video may not significantly impact the generated captions as our deployed image caption models generate English descriptions based purely on input frames. However, in terms of video distributions, there may exist differences (such as in behaviors, activities, and events) between videos stemming from different countries due to varied cultural backgrounds.
>
> To examine this hypothesis, we selected two 2 million subsets of InternVid: one consisting of only English videos (InternVid-2M-EN) and another with only Chinese videos (InternVid-2M-CN). It's important to note that whether the videos are in English or Chinese, we generate captions in English.
>
> Our ViCLIP-B model was pretrained on these subsets, and we conducted zero-shot experiments as described below. Due to resource constraints, we trained the ViCLIP-B with a batchsize of 4096 using 8 A100 GPUs with a mask ratio set to 0.9. The remaining training settings were kept consistent with those outlined in the paper.
>
> We found that the model pretrained with InternVid-2M-EN outperformed that with InternVid-2M-CN notably in both zero-shot action recognition on K400/600/700 and video retrieval. This result can be attributed to the fact that InternVid-2M-EN has a data distribution much closer to downstream task data than InternVid-2M-CN, as all used task videos are sourced from English sources.
>
> ### Table B. Zero-shot action recognition results of ViCLIP using different pretraining sources on Kinetics 400/600/700.
> |Method | Training Data | K400 |  | K600 |  | K700 |  |
> |:---:|:---:|:---:|:---:|:---:|:---:|:---:|:---:|
> | | |top-1 | AVG | top-1 | AVG | top-1 | AVG |
> |ViCLIP | +InternVid-2M-EN | 40.20 | 53.68 | 37.40 | 51.00 | 29.60 | 42.06 |
> |ViCLIP | +InternVid-2M-CN | 35.9 | 49.73 | 33.70 | 47.05 | 26.90 | 39.02 |
>
> ### Table C. Results of zero-shot video retrieval of ViCLIP using different pretraining sources on MSR-VTT, LSMDC, DiDeMo, MSVD, and ActivityNet.
> |Method | Data | MSR-VTT | | LSMDC | | DiDeMo | | MSVD | | ANet | |
> |:---:|:---:|:---:|:---:|:---:|:---:|:---:|:---:|:---:|:---:|:---:|:---:|
> | | |T2V | V2T | T2V | V2T | T2V | V2T | T2V | V2T | T2V | V2T |
> |ViCLIP | +InternVid-2M-EN | 24.1 | 24.1 | 9.8 | 9.1 | 10.3 | 15.6 | 31.4 | 50.5 | 7.1 | 12.0 |
> |ViCLIP | +InternVid-2M-CN | 22.2 | 22.2 | 8.4 | 8.7 | 9.9 | 15.1 | 29.6 | 48.5 | 5.7 | 9.7 |
>
> **Q5: Minor comments on Figure 2. Similar colors like green and dark green are used, but this may cause confusion. Using more distinct and discrete colors, such as red and blue, could enhance the clarity.**
>
> Sure. We will update Figure 2 and check other ones for better clarity.
>
> **Q6: Consideration with Table 6. Comment of VideoCrafter, VideoFusion is absent. Does InternVID also improve VideoCrafter, VideoFusion in IS, FID, FVD, and CLIPSIM? Can you provide further analysis about the choice of InternVID-Aes-18M rather than exploiting the full InternVID-200M?**
>
> The lack of full training code and details from VideoCrafter and VideoFusion poses a challenge for reproducing their results and further training with additional data.
>
> The decision to use InternVID-Aes-18M instead of the full InternVID-200M is driven by techniques from studies [R4]. These studies suggested that image aesthetic quality significantly impacts text-to-image generation, even more than the sheer quantity of images. We hypothesize that this principle can be extrapolated to video generation as well. Therefore, we decided to curate a subset based on video aesthetics for text-to-video generation.
>
> The process to compute video aesthetics involved calculating the image aesthetic scores of four randomly sampled frames from each video clip. The tool used for this task was the LAION aesthetic predictor. The highest score among the four frames was then chosen to represent the aesthetic score of that particular video.
>
> This approach ensured that our model was trained on aesthetically pleasing videos, potentially improving the overall quality of the generated outputs. However, note that this approach has not been explicitly tested on VideoCrafter or VideoFusion. Therefore, it remains uncertain whether InternVid would result in improved IS, FID, FVD, and CLIPSIM metrics for these models. Further experiments would be needed to confirm this hypothesis.
>
> [R4] Emu: Enhancing Image Generation Models Using Photogenic Needles in a Haystack. In arxiv. 2023.

---

> > ### Author Response · Authors · 2023-11-20
> > **Response to Official Review of Submission4917 by Reviewer 1UqW: Q7**
> >
> > **Q7: Consideration with Section 5.2 and Figure 13~17. To insist that InternVID provides more powerful video-text aligned representations for video-centric dialogue systems, authors must compare the results from VideoChat-ViCLIP with that of vanilla VideoChat. Can you provide quantitative and qualitative comparisons, showing improvement of the results after replacement of the previous visual encoder with ViCLIP?**
> >
> > In terms of quantitative comparison, as shown in Table D, VideoChat-ViCLIP significantly outperforms the vanilla VideoChat (using Eva-g as the vision encoder) and other systems across all evaluation aspects of the quantitative video conversation evaluation framework in [R5]. Specifically, the model shows remarkable improvements in the correctness of information (from 2.23 to 2.86), contextual understanding (from 2.53 to 3.08), and temporal understanding (from 1.94 to 2.36). The average score also increases from 2.29 to 2.64, showing an overall performance gain.
> >
> > For qualitative comparison, it is given in the [anonymous link](https://s1.imagehub.cc/images/2023/11/17/b164d37b037b2fcf26a2bc5d6947e3ae.jpeg). It is important to note that with the replacement of the previous visual encoder with ViCLIP, the system has become more adept at understanding and generating responses that are contextually appropriate, contain correct information, and accurately reflect the temporal dynamics of the video content. This leads to more coherent and meaningful dialogue exchanges in a video-centric dialogue system.
> >
> > Both quantitative and qualitative comparisons validate that the video-text aligned representations in ViCLIP are effective for video-centric dialogue systems.
> >
> > ### Table D. Performance benchmarking of text generation models.
> > |              Evaluation   Aspect | Correctness of Information | Detail Orientation | Contextual Understanding | Temporal Understanding | Consistency | Avg  |
> > |:--------------------------------:|----------------------------|--------------------|--------------------------|------------------------|-------------|------|
> > |            VideoChat (Eva-g)     |            2.23            |         2.5        |           2.53           |          1.94          |     2.24    | 2.29 |
> > |           LLaMA Adapter          |            2.03            |        2.32        |            2.3           |          1.98          |     2.15    | 2.16 |
> > |            Video LLaMA           |            1.96            |        2.18        |           2.16           |          1.82          |     1.79    | 1.98 |
> > |           Video-ChatGPT          |             2.4            |        2.52        |           2.62           |          1.98          |     2.37    | 2.38 |
> > |         VideoChat-ViCLIP-L         |            2.86            |        2.52        |           3.08           |          2.36        |     2.4     | 2.64 |
> >
> > [R5] Video-ChatGPT: Towards Detailed Video Understanding via Large Vision and Language Models. In arxiv. 2023.

---

> > > ### Comment · Reviewer_1UqW · 2023-11-21
> > >
> > > Thank you for your detailed answers. We appreciate the answers regarding the comparison between T5 and Llama, further analysis between InternVid-2M-EN and InternVid-2M-CN considering language/ content diversity, and quantitative comparison between vanilla VideoChat and VideoChat-ViCLIP. I would like to see the inclusion of these analyses in the final version, which resolved our main concerns, so we raise the score to 6.

---

> > > > ### Author Response · Authors · 2023-11-21
> > > > **Response to Official Review of Submission4917 by Reviewer 1UqW**
> > > >
> > > > We sincerely appreciate your time and effort in reviewing our work, as well as the constructive feedback you have provided throughout this process. We are delighted to know that our responses were able to address your main concerns effectively.
> > > >
> > > > We understand the importance of the additional analyses and comparison between T5 and Llama, InternVid-2M-EN and InternVid-2M-CN considering language/content diversity, and the quantitative comparison between vanilla VideoChat and VideoChat-ViCLIP. Therefore, we assure you that these detailed components will be included in the final version of our manuscript.
> > > >
> > > > Your insightful comments have undoubtedly helped us improve the quality of our work, and we are grateful for your positive interaction during this review process. With your encouraging score, we feel motivated to make the necessary adjustments to enhance the overall quality of our paper.
> > > >
> > > > Once again, thank you for your valuable contribution during the review period, and we look forward to incorporating your suggestions into our final document.

---

### Official Review · Reviewer_6e9v · 2023-10-30

**Soundness:** 3 good
**Presentation:** 3 good
**Contribution:** 3 good
**Rating:** 8
**Confidence:** 5

**Summary:**

- This paper introduces a web-scale video language dataset InternVid comprised of 234M clips and 760K hours.
- It introduces ViCLIP, a video-language model based on ViT-L, pretrained with contrastive learning (like CLIP) and masked autoencoder (like VideoMAE, MAE).
- This work expands the utility of their dataset to video understanding tasks like recognition and retrieval, and video generation.

**Strengths:**

- It introduces a web-scale video-language dataset bridging the gap of lack of such datasets, unlike image-text domains.
- It evaluates on multiple benchmarks in both finetuned and zero-shot setups.
- The dataset curation is fairly detailed and I also like the hierarchical video caption generation strategy.

**Weaknesses:**

- The majority of the comparisons are with CLIP which was pretrained with image-text pairs and not video-text; I would like to see how ViCLIP performs compared to other video or video-language models (e.g., pretrained on HD-VILA, HT100M, YT100M), I think that would be a fair comparison than comparing with image-text models.
- I would encourage authors to dig deeper and find a more concrete argument/explanation: why does pretraining with InternVid-10M-FLT or InternVid-10M-DIV perform better in **zero-shot** than InternVid-200M, but not when **finetuned**.
- In fig. 7 and 8, the experiments are done only on scaling the dataset size, it would be interesting to see the effect of model scaling in addition to the dataset scaling. I would encourage you to add such experiments in the final version.
- I would be interested to see linear evaluation (i.e., a single FC layer, or use linear SVM) performance on the downstream benchmarks.

**Questions:**

- Will you share the pretrained and finetuned models with the supporting code base (e.g., data processing, pretraining, finetuning, generation)?
- Could you please share the processed clips, even processing the data by individuals (typically for academic researchers) would be a difficult task considering its massive size. Additionally, we all know the unavailability of videos due to location constraints, permission issues, etc., so even if the full data is not possible to share at least share the 3 10M versions.
- I suggest releasing the fixed embeddings of the datasets from the trained (e.g., pretrained, finetuned) models.
- Did you investigate if the InternVid has any sort of bias in the curated clips, could you please share a report with such details? Bias could be of many forms e.g., location/race/gender per action category. A suggested reference: https://arxiv.org/abs/1505.01257
- Did you investigate, if ViCLIP is robust against some of the OOD setups, some of the popular benchmarks are Mimetics, RareAct etc. For more details please see: https://arxiv.org/abs/2306.02014

---

> ### Author Response · Authors · 2023-11-16
> **Response to Official Review of Submission4917 by Reviewer 6e9v - Q1 & Q2**
>
> **Q1: The majority of the comparisons are with CLIP which was pretrained with image-text pairs and not video-text; I would like to see how ViCLIP performs compared to other video or video-language models (e.g., pretrained on HD-VILA, HT100M, YT100M), I think that would be a fair comparison than comparing with image-text models.**
>
> We understand your concern about the comparison with CLIP, which was pretrained on image-text pairs. Indeed, to provide a more fair comparison, we included results against CLIP-ViP, a version of CLIP pretrained on HD-VILA with coarse synthetic captions (one frame caption used as the video caption) and ASR text. However, please note that this comparison still has limitations as CLIP-ViP only offers a ViT-base version and some of its variants are trained with different settings.
>
> Turning our attention to Table A, when evaluated on MSR-VTT, our model, ViCLIP-B, trained on InternVid-200M/-10M-FLT outperforms CLIP-ViP-B, trained on HD-VILA-100M with subtitles, achieving a higher R@1 score (50.7%/49.0% vs. 47.7%). CLIP-ViP-B reaches similar performance (49.6%) only when using a combination of captions and subtitles from HD-VILA-100M. It's worth noting that CLIP-ViP-B uses more frames for pretraining than ours (12 vs. 8).
>
> ### Table A. Fine-tuned video retrieval on MSR-VTT.
> |Method | Data | #Frames in Train | T2V | V2T |
> |:---:|:---:|:---:|:---:|:---:|
> |CLIP-ViP-B | +HD-VILA-100M | 12 | 47.7 |  |
> |CLIP-ViP-B | +HD-VILA-100M(sub+cap) | 12 | 49.6 |  |
> |CLIP-ViP-B | +HD-VILA-100M(sub)+Im-text data | 12 | 49.1 |  |
> |ViCLIP-B | +InternVid-200M | 8 | 50.7 | 49.4 |
> |ViCLIP-B | +InternVid-10M-FLT | 8 | 49.0 | 49.2 |
>
>
> **Q2: I would encourage authors to dig deeper and find a more concrete argument/explanation: why does pretraining with InternVid-10M-FLT or InternVid-10M-DIV perform better in zero-shot than InternVid-200M, but not when finetuned.**
>
> The varying performance of ViCLIP (trained on InternVid-10M-FLT/DIV and InternVid-200M) in zero-shot vs. fine-tuned action recognition can be attributed to the differing demands of these tasks. Zero-shot action recognition primarily depends on the alignment between video and text representations, while fine-tuned action recognition focuses on discriminating video representation as we fine-tune the video encoder using supervised video-label data and discard the text encoder.
> So it's important to note that there's no theoretical guarantee for correlation between a model's zero-shot and fine-tuned capabilities.
>
> You correctly noted that InternVid-10M-DIV/FLT performs better in zero-shot action recognition (Table 2), but not in the fine-tuned setting (Table 3), when compared to InternVid-200M. Our interpretation is as follows:
> 1. ViCLIP trained on InternVid-200M likely has a more robust video representation due to a larger quantity and diversity of videos.
> 2. On the other hand, ViCLIP trained on InternVid-10M-FLT/DIV might excel in video-text alignment—made possible by limiting the number of clips drawn from identical videos. This results in more diverse training data compared to InternVid-200M’s, thereby facilitating contrastive learning, enhancing video-text representation, and driving better outcomes for zero-shot action recognition and tasks like retrieval.

---

> > ### Author Response · Authors · 2023-11-16
> > **Response to Official Review of Submission4917 by Reviewer 6e9v - Q3 & Q4**
> >
> > **Q3: In fig. 7 and 8, the experiments are done only on scaling the dataset size, it would be interesting to see the effect of model scaling in addition to the dataset scaling. I would encourage you to add such experiments in the final version.**
> >
> > Scaling the model size is indeed a computationally demanding process. However, we provide a comparison between two versions of ViCLIP, -L (300M) and -B (80M), when trained on InternVid in Tables B, C, and D. These tables distinctly demonstrate that moving from the base to large model, ViCLIP's zero-shot and fine-tuned video retrieval performance, as well as zero-shot action recognition, can be consistently improved. Training larger models than these was unfortunately unfeasible within our rebuttal timeline.
> >
> > ### Table B. Scaling model in fine-tuned video retrieval on MSR-VTT, LSMDC, DiDeMo, MSVD, and ActivityNet
> > |Method | Data | MSR-VTT | | LSMDC | | DiDeMo | | MSVD | | ANet | |
> > |:---:|:---:|:---:|:---:|:---:|:---:|:---:|:---:|:---:|:---:|:---:|:---:|
> > | | |T2V | V2T | T2V | V2T | T2V | V2T | T2V | V2T | T2V | V2T |
> > |ViCLIP-B | +InternVid-200M | 50.7 | 49.4 | 25.3 | 25.4 | 41.1 | 40.8 | 69.0 | 69.3 | 37.7 | 35.8 |
> > |ViCLIP-L | +InternVid-200M | 53.7 | 53.4 | 29.3 | 29.3 | 51.1 | 50.8 | 79.9 | 78.4 | 52.8 | 51.1 |
> > |ViCLIP-B | +InternVid-10M-FLT | 49.0 | 49.2 | 24.4 | 23.7 | 40.0 | 41.4 | 72.2 | 73.7 | 38.3 | 37.0 |
> > |ViCLIP-L | +InternVid-10M-FLT | 52.5 | 51.8 | 33.0 | 33.0 | 49.4 | 50.2 | 77.2 | 79.0 | 49.8 | 48.1 |
> >
> >
> > ### Table C. Scaling model in zero-shot video retrieval on MSR-VTT, LSMDC, DiDeMo, MSVD, and ActivityNet.
> > |Method | Data | MSR-VTT | | LSMDC | | DiDeMo | | MSVD | | ANet | |
> > |:---:|:---:|:---:|:---:|:---:|:---:|:---:|:---:|:---:|:---:|:---:|:---:|
> > | | |T2V | V2T | T2V | V2T | T2V | V2T | T2V | V2T | T2V | V2T |
> > |ViCLIP-B | +InternVid-200M | 37.4 | 36.1 | 16.5 | 15.2 | 16.6 | 22.6 | 44.3 | 67.0 | 13.3 | 21.7 |
> > |ViCLIP-L | +InternVid-200M | 39.3 | 39.5 | 18.3 | 16.6 | 17.1 | 25.5 | 47.3 | 70.0 | 13.7 | 21.6 |
> > |ViCLIP-B | +InternVid-10M-FLT | 38.6 | 38.5 | 18.5 | 17.0 | 16.3 | 25.0 | 44.8 | 67.2 | 13.0 | 21.8 |
> > |ViCLIP-L | +InternVid-10M-FLT | 42.4 | 41.3 | 20.1 | 16.9 | 18.4 | 27.9 | 49.1 | 75.1 | 15.1 | 24.0 |
> >
> >
> > ### Table D. Scaling model in zero-shot action recognition results on Kinetics 400/600/700.
> > |Method | Training Data | K400 |  | K600 |  | K700 |  |
> > |:---:|:---:|:---:|:---:|:---:|:---:|:---:|:---:|
> > | | |top-1 | AVG | top-1 | AVG | top-1 | AVG |
> > |ViCLIP-B | +InternVid-200M | 56.58 | 69.20 | 53.57 | 66.20 | 45.82 | 58.28 |
> > |ViCLIP-L | +InternVid-200M | 59.80 | 71.09 | 57.80 | 69.34 | 49.30 | 61.25 |
> > |ViCLIP-B | +InternVid-10M-FLT | 58.52 | 71.11 | 55.37 | 68.27 | 47.09 | 59.98 |
> > |ViCLIP-L | +InternVid-10M-FLT | 64.80 | 75.70 | 62.20 | 73.53 | 54.30 | 66.38 |
> >
> > These tables clearly demonstrate the benefits of model scaling, and we aim to explore this area further in future work as resources permit.
> >
> >
> > **Q4: I would be interested to see linear evaluation (i.e., a single FC layer, or use linear SVM) performance on the downstream benchmarks.**
> >
> > We appreciate your interest in linear evaluation performance. In response, we present the linear action recognition results on Kinetics-400 in Table E below. It's noteworthy that ViCLIP, trained on InternVid-10M-FLT/-200M, delivers a much higher top-1 accuracy compared to when trained on WebVid-10M (with a more than 10-point increase), mirroring our findings in fine-tuned action recognition settings.
> >
> > Comparing with other approaches, ViCLIP-L offers performance close to TVTSv2-H/-B, which incorporate extra learnable parameters for spatiotemporal modeling. Moreover, it significantly outperforms VideoMAEv2-H. This result can be attributed to the fact that MAE-based methodologies generally underperform in linear evaluations.
> >
> > ### Table E. Linear action recognition results on Kinetics-400.
> > |Method | Pretraining Data | Kinetics-400 | |
> > |:---:|:---:|:---:|:---:|
> > | | |top-1 | top-5 |
> > |VideoMAE-B [R1]| Kinetics-400 | 20.4 | - |
> > |VideoMAEv2-H [R2]| Kinetics+SthSth+AVA+WebVid2M | 25.8 | - |
> > |TVTS-B [R3]| +YT-Temporal-180M | 60.8 | - |
> > |TVTSv2-B [R4]| +YT-Temporal-180M+WebVid-2M | 70.1 | - |
> > |TVTSv2-H [R4]| +YT-Temporal-180M+WebVid-2M | 73.1 | - |
> > |ViCLIP-L| +WebVid-10M | 60.0 | 82.9 |
> > |ViCLIP-L| +InternVid-10M-FLT | 71.1 | 90.4 |
> > |ViCLIP-L| +InternVid-200M | 71.7 | 90.9 |
> >
> > [R1] Videomae: Masked autoencoders are data-efficient learners for self-supervised video pre-training. In NeurIPS. 2022.
> >
> > [R2] Videomae v2: Scaling video masked autoencoders with dual masking. In CVPR. 2023.
> >
> > [R3] Learning Transferable Spatiotemporal Representations from Natural Script Knowledge. In CVPR. 2023
> >
> > [R4] TVTSv2: Learning Out-of-the-box Spatiotemporal Visual Representations at Scale. In arXiv. 2023.

---

> > > ### Author Response · Authors · 2023-11-16
> > > **Response to Official Review of Submission4917 by Reviewer 6e9v - Q5-Q9**
> > >
> > > **Q5: Will you share the pretrained and finetuned models with the supporting code base (e.g., data processing, pretraining, finetuning, generation)?**
> > >
> > > We are committed to the principle of transparency and openness in scientific research. As such, we pledge to release all our mentioned models and code bases upon acceptance of our work.
> > >
> > > **Q6: Could you please share the processed clips, even processing the data by individuals (typically for academic researchers) would be a difficult task considering its massive size. Additionally, we all know the unavailability of videos due to location constraints, permission issues, etc., so even if the full data is not possible to share at least share the 3 10M versions.**
> > >
> > > YouTube's policies prohibit institutions or researchers from releasing data (whether original or processed) that has been crawled from its platform. We strictly adhere to these regulations in all our operations.
> > >
> > > Despite this, we have faith in our vibrant and dedicated community. We anticipate that the processed datasets of varying sizes mentioned will soon be shared anonymously on the internet. This is because our community is filled with volunteers who are eager to contribute and have a strong sense of collective spirit.
> > >
> > > **Q7: I suggest releasing the fixed embeddings of the datasets from the trained (e.g., pretrained, finetuned) models.**
> > >
> > > We are working on it. They will be released step by step, starting from a subset of InternVid first, e.g. InternVid-10M-FLT.
> > >
> > > **Q8: Did you investigate if the InternVid has any sort of bias in the curated clips, could you please share a report with such details? Bias could be of many forms e.g., location/race/gender per action category.**
> > >
> > > Yes, we have conducted an investigation into potential biases in the InternVid dataset. Specifically, we focused on age, gender, and race distributions, as these are commonly recognized areas where bias can occur. Our methodology consisted of counting keywords related to these categories in the generated video captions.
> > >
> > > It's important to note that these synthetic captions may not fully reflect the truth of the corresponding videos, thereby creating a gap between our analysis and the actual reality.Here are the results of our analysis:
> > >
> > > 1. Age distribution: We counted nouns related to children, grown-ups, and the elderly. We found that 30.71% of the video captions contained such descriptions. Within this subset, the majority were about adults (84.59%), followed by children (15.31%) and barely any mentions of senior citizens (0.08%).
> > >
> > > 2. Gender distribution: We counted nouns specifically related to males and females. According to our findings, 33.7% of video captions contained some form of gender-related text. Among these, 64.27% pertained to men and 35.73% pertained to women.
> > >
> > > 3. Race distribution: Only around 2.51% of video captions contained descriptions related to race. This could be due to the limitations of our captioning pipeline, which might not be capable of capturing such attributes accurately. Further exploration using a dedicated race recognition model is needed for more accurate statistics.
> > >
> > > We intend to add this analysis to our final report and include relevant references, such as the one you've provided along with others. This is an initial investigation, and further work is required for a comprehensive understanding of potential biases in the InternVid dataset.
> > >
> > > **Q9: Did you investigate, if ViCLIP is robust against some of the OOD setups, some of the popular benchmarks are Mimetics, RareAct etc. For more details please see:https://arxiv.org/abs/2306.02014**
> > >
> > > We did not conduct specific investigations related to Out-Of-Distribution (OOD) performance in this version of the paper. However, we speculate that our results on zero-shot action recognition and video retrieval could partially highlight ViCLIP's performance under OOD conditions, given the evident distribution gap between the pretraining data and evaluation data.
> > >
> > > In light of the feedback, we intend to incorporate an analysis focused on OOD performance in the final version of this work. We appreciate the valuable reference shared for evaluating video model capabilities from a perspective centered around distribution shifts, instead of solely focusing on common video task performance. This approach indeed presents unique insights and offers an invigorating perspective on model evaluation.
> > > The reference will be thoroughly studied, its key findings will be integrated into our analysis, and it will be duly cited in our revised manuscript. Furthermore, we will also elaborate on the relevant work in relation to OOD evaluation to provide a comprehensive understanding of the topic.
> > >
> > > Our goal with these improvements is to present a more robust analysis of ViCLIP's performance, particularly under OOD conditions. We believe that such enhancements will significantly enrich the content of our paper.

---

> ### Comment · Reviewer_6e9v · 2023-11-17
> **Post rebuttal comment**
>
> I would like to thank the authors for responding to my comments. I hope the authors would make the said changes in their final version as promised. I am happy to stick to my original rating in favor of acceptance.

---

> > ### Author Response · Authors · 2023-11-20
> > **Response to post rebuttal comment of Submission4917 by Reviewer 6e9v**
> >
> > We greatly appreciate your time and insightful comments throughout the review process. We are glad to hear that you found our responses satisfactory.
> >
> > As previously mentioned, we fully commit to implementing all the proposed changes in the final manuscript. Your feedback has immensely helped us in improving the quality of our research work.
> >
> > Once again, thank you for your positive rating and for advocating for the acceptance of our paper. We look forward to the opportunity to share the final version of our work with the wider community.

---

### Official Review · Reviewer_G48V · 2023-10-31

**Soundness:** 2 fair
**Presentation:** 3 good
**Contribution:** 3 good
**Rating:** 6
**Confidence:** 4

**Summary:**

This paper introduces a new video and language dataset, INTERNVID, which includes 234M video-text pairs lasting 760K hours. In addition to the dataset, this paper also introduce a baseline model, ViCLIP, demonstrating its performance on various downstream applications after pre-training on INTERNVID dataset.

**Strengths:**

1. The paper is well written, and each section is easy to follow.

2. The proposed large-scaled video-text dataset can contribute to the community, scaling up the current model and largely improving the video and language feature representation learning.

3. This paper provides very detailed statistic for the dataset, and also reveal the method for data curation.

4. By leveraging the new dataset, this paper demonstrates extensive experiments on many downstream applications and achieving promising results on many benchmarks.

**Weaknesses:**

1. There are not many analyses and design justification for the proposed ViCLIP. If ViCLIP is claimed as one of the contributions of this paper. The proposed architecture is not novel and the masking idea needs further justification. For example, the efficiency gain vs. the performance drop, and the ablation over the masking ratio.

2. The video caption is generated by language model from frame-level captions. In this case, will this reduce the number of motion-related words that need to be captured from video-based understanding?

**Questions:**

Given 10M pretrained data, the ViCLIP receives better zero-shot action recognition results from WebVid and better fine-tune action recognition results in Table 2 and 3. And the gain from 50M, 200M INTERNVID pretraining is minor. Does it mean the pretraining data is not the more the better? The performance of vision and language model will be saturated when the pretraining data reach to a certain scale? Could you please provide more insights here?

---

> ### Author Response · Authors · 2023-11-14
> **Response to Official Review of Submission4917 by Reviewer G48V**
>
> **Q1: There are not many analyses and design justifications for the proposed ViCLIP. If ViCLIP is claimed as one of the contributions of this paper. The proposed architecture is not novel and the masking idea needs further justification. For example, the efficiency gain vs. the performance drop, and the ablation over the masking ratio.**
>
> The primary aim of applying ViCLIP on InternVid is to validate the efficacy of our proposed dataset in video understanding tasks. We concur that the architecture and training methodology for ViCLIP are not novel; it follows the design principles of CLIP, with the addition of mask modeling for training efficiency.
>
> The masking scheme, specifically the choice of a 0.9 masking ratio from VideoMAE, was adopted to balance computational efficiency and performance. Below, we present Table A, which lists the trade-off between efficiency gains and performance drops at different masking ratios for ViCLIP-B. As can be seen, decreasing the masking ratio improves performance but increases GPU memory usage significantly. The masking ratio of 0.9 was chosen because it provides the highest efficiency with a tolerable performance drop on downstream tasks.
>
> - Table A. Fine-tuned video retrieval on MSR-VTT from ViCLIP with different masking ratios. ``Mem" denotes single GPU memory usage with per GPU batch size 128.
> |Method | Data | Masking ratio | T2V | Mem/G|
> |:---:|:---:|:---:|:---:|:---:|
> |ViCLIP-B | +InternVid-10M | 0.7 | 48.0 | 27.9 |
> |ViCLIP-B | +InternVid-10M | 0.8 | 47.7 | 19.2 |
> |ViCLIP-B | +InternVid-10M | 0.9 | 47.4 | 12.1 |
>
> **Q2: The video caption is generated by language model from frame-level captions. In this case, will this reduce the number of motion-related words that need to be captured from video-based understanding?**
>
> From a statistical perspective, generating video captions from frame-level captions using a language model has a **negligible** effect on the number of motion-related words captured for video-based understanding.
>
> To illustrate this, we counted the unique verbs (using nltk package) in the captions from a 10m subset of InternVid under two settings:
>
> 1. In the first setting, the captions are video captions generated by the language model.
>
> 2. In the second setting, the captions are frame-wise ones from BLIP2 and tag2text.
>
> We found that the number of unique verbs in the video captions is **109,859**, whereas for the frame-wise captions it is slightly higher at **109,895**. This **small discrepancy** suggests that almost no motion-related words are lost during the caption generation process by LM. Therefore, we believe our approach maintains most of the important motion-related information needed for video understanding.
>
> **Q3: Given 10M pretrained data, the ViCLIP receives better zero-shot action recognition results from WebVid and better fine-tune action recognition results in Table 2 and 3. And the gain from 50M, 200M INTERNVID pretraining is minor. Does it mean the pretraining data is not the more the better? The performance of vision and language model will be saturated when the pretraining data reach to a certain scale? Could you please provide more insights here?**
>
> For the subset of 10M pretraining data, ViCLIP performs better in both zero-shot and fine-tuned action recognition when utilized with InternVid-10M-FLT/DIV compared to WebVid. This performance differences between models using InternVid-10M-FLT/DIV and InternVid-10M underscores the importance of sampling unique clips from videos - a hypothesis supported by our initial explorations.
>
> Our findings from Tables 2 and 5 along with Figures 7 and 8 demonstrate a consistent performance improvement in zero-shot action recognition and fine-tuned video retrieval as the pretraining data size escalates. However, the enhancement in zero-shot video retrieval is only marginal, which we believe stems from the limited increase in video-text diversity during pretraining. In particular, the more clips sampled from the same video, the higher tendency they have to share similar semantics. Furthermore, while a generative captioning method aids in video descriptions, the limited capacity of the captioning model restricts the diversity of produced captions, making them somewhat inferior to human annotations. These findings motivate us to explore scaling InternVid-10M-FLT/DIV and improving video captions through advanced captioning methods, language modeling, and human feedback techniques.
>
> From this standpoint, our current experimental data does not conclusively suggest a saturation point for vision and language model performance as pretraining data scales up. We can only affirm that zero-shot video retrieval performance will reach a plateau at a certain scale of pretraining data when further diversification of the data used becomes unavailable.

---

> ### Comment · Reviewer_G48V · 2023-11-21
> **Post rebuttal comment**
>
> Thanks all authors for their effort in the rebuttal.
>
> The rebuttal address most of my concerns, so I keep my rating suggesting acceptance of this work.
>
> Thanks

---

### Official Review · Reviewer_gWXX · 2023-11-01

**Soundness:** 3 good
**Presentation:** 2 fair
**Contribution:** 3 good
**Rating:** 8
**Confidence:** 5

**Summary:**

This paper proposes a large-scale video-text dataset for video representation learning. Considering the noise and low correlation in ASR transcripts, this paper utilizes captioning models and a pre-trained language model to generate video descriptions. The authors pre-train the ViCLIP model on the collected dataset and the model shows strong performance on action recognition and text-video retrieval tasks. They also explore potential applications of this dataset on text-to-video generation and video-centric dialogue systems.

**Strengths:**

- The dataset is large-scale and diverse, and the generated descriptions have better relevance with videos.
- The pre-trained model shows good performance.
- The experiments are extensive.

**Weaknesses:**

- Some details are not presented clearly. 1) How many descriptions are generated at the coarser level? Are all descriptions used for training the final model at the finer level? 2) Which pre-trained language model is used for processing captions? 3) The details about DIV and FLT are not introduced.
- A closely related work, CLIP-ViP[R1], is not included in related works and compared. It also adopts a caption model to generate descriptions for video clips. It is corresponding to the coarser level in this paper.
- The performance of InterVid-10M-DIV, InterVid-10M-FLT is weird. It shows much better performance in zero-shot action recognition in Table 2, but poor in the fine-tuned setting of Table 3. The authors give reasons for false negatives, but what is the training batch size, given the much larger training data, I think the possibility of the same video clip appearing in the same batch is low.

[R1]: CLIP-ViP: Adapting Pre-trained Image-Text Model to Video-Language Representation Alignment, ICLR 2023.

**Questions:**

Refering to the weaknesses.

---

> ### Author Response · Authors · 2023-11-14
> **Response to Official Review of Submission4917 by Reviewer gWXX - Q1 & Q2**
>
> **Q1: Some details are not presented clearly. 1) How many descriptions are generated at the coarser level? Are all descriptions used for training the final model at the finer level? 2) Which pre-trained language model is used for processing captions? 3) The details about DIV and FLT are not introduced.**
>
> 1) We generated both coarse and fine descriptions for **all** of the collected and segmented video clips, totaling 234 million. However, not all descriptions at the finer level were used in training the final model. For our largest experimental setting conducted on InternVid-200M, we trained ViCILP on a random selection of 200 million clips with synthetic video captions derived from both coarse and fine-level captions.
>
> 2) The language model predominantly used for processing captions was T5-summary, accessible at [this URL](https://huggingface.co/mrm8488/flan-t5-large-finetuned-openai-summarize_from_feedback). A small proportion of conversions were handled by Vicuna. We found that, when compared to other options like LLM (7b or 13b), T5 offered significantly faster processing speeds and did not generate hallucinations. On average, T5 completed each summary within 14.96ms while Llama2-7b took longer at 81.51ms, both using A100-80G.
>
> 3) For DIV (diversity sampling), we aimed to sample video clips from all long videos available to maximize data diversity. This was done by counting the frequencies of long videos in the segmented clip pool and sampling clips with probabilities inverse to these frequencies. Here is a pseudocode example of this process:
>
> ```python
>     from collections import Counter
>     import json
>     import random
>     import numpy as np
>
>     data = json.load(open("/path/to/to_sample"))
>     video_id = set([x["video"].split("/")[-1][:11] for x in data])
>     video_id_counter = Counter([x["video"].split("/")[-1][:11] for x in data])
>     sampling_weights = [1.0 / video_id_counter[x["video"].split("/")[-1][:11]] for x in data]
>     np.random.seed(42)
>     sampling_weights = np.array(sampling_weights)
>     sampling_weights = sampling_weights / sampling_weights.sum()
>     sampled_index = np.random.choice(len(data), 10647458, replace=False, p=sampling_weights)
>     data = [data[i] for i in sampled_index]
>     json.dump(data, open("/path/to/sampled", "w"))
> ```
>
> For FLT (filtering), we applied a series of filtering strategies to video data alongside DIV sampling. These included:
>
> a) Removing video clips shorter than 1s (approximately 23.15% of the total) or longer than 120s (around 0.84% of the total).
>
> b) Computing CLIPScore for each video clip using a randomly sampled frame from the clip with OpenAI's CLIP-ViT-L/14, then selecting clips within the top 30% of CLIPScores.
>
> c) Sampling 10M out of the remaining clips using DIV sampling.
>
> **Q2: A closely related work, CLIP-ViP[R1], is not included in related works and compared. It also adopts a caption model to generate descriptions for video clips. It is corresponding to the coarser level in this paper.**
>
> We acknowledge the omission of CLIP-ViP[R1] in the related works. This will be rectified in the final manuscript where a thorough discussion on CLIP-ViP and its relation to our work will be included.
>
> In response to your request for a comparison with CLIP-ViP, we have provided preliminary results in Table X. Please note, these results are not strictly like-for-like due to differing parameters such as number of sampled frames during training, training epochs, and resource constraints that limited our ability to retrain all models.
>
> As seen in Table A, when evaluated on MSR-VTT, our model (ViCLIP-B used with InternVid-200M/-10M-FLT) outperforms CLIP-ViP-B used with HD-VILA-100M, giving a better R@1 score (50.7%/49.0% vs. 47.7%). However, when captions and subtitles from HD-VILA-100M are combined, CLIP-ViP-B achieves comparable result of 49.6%. An important note is that CLIP-ViP-B uses more frames for pretraining than ours (12 vs. 8).
>
> This suggests that synthetic video captions from our InternVid dataset can compete effectively against HD-VILA-100M's combined subtitles and captions.
>
> - Table A. Fine-tuned video retrieval on MSR-VTT.
> |Method | Data | #Frames in Train | T2V | V2T |
> |:---:|:---:|:---:|:---:|:---:|
> |CLIP-ViP-B | +HD-VILA-100M | 12 | 47.7 |  |
> |CLIP-ViP-B | +HD-VILA-100M(sub+cap) | 12 | 49.6 |  |
> |CLIP-ViP-B | +HD-VILA-100M(sub)+Im-text data | 12 | 49.1 |  |
> |ViCLIP-B | +InternVid-200M | 8 | 50.7 | 49.4 |
> |ViCLIP-B | +InternVid-10M-FLT | 8 | 49.0 | 49.2 |
>
> [R1]: CLIP-ViP: Adapting Pre-trained Image-Text Model to Video-Language Representation Alignment, ICLR 2023.

---

> > ### Comment · Reviewer_gWXX · 2023-11-15
> >
> > Thank you for addressing my concerns; I have gained some clarity on certain concerns. The use of coarse-level captions indeed seems similar to that in CLIP-ViP, but the dataset that provides fine-level captions appears to be a significant contribution. However, I have some questions:
> > 1. What is the mixing strategy for coarse and fine-level captions when training InternVid-200M? What motivated the inclusion of coarse-level captions? I would like to see an ablation study that uses coarse-level captions, fine-level captions, and both, possibly at a scale of 10M.
> > 2. I would appreciate more examples from the dataset, especially those that illustrate the contrast between coarse and fine-level captions.
> > 3. Regarding the T5-summary and Vicuna models mentioned, were they fine-tuned for processing captions? I suspect that these models are not capable enough for zero-shot handling of captions. Perhaps GPT-3.5-Turbo or GPT-4 might be suitable.

---

> > > ### Author Response · Authors · 2023-11-20
> > > **Response to Official Comment by Reviewer gWXX: Q1 & Q2**
> > >
> > > **Q1: What is the mixing strategy for coarse and fine-level captions when training InternVid-200M? What motivated the inclusion of coarse-level captions? I would like to see an ablation study that uses coarse-level captions, fine-level captions, and both, possibly at a scale of 10M.**
> > >
> > > As given in Figure 2 and Section 3.2, we only explored and experimented with one setting that summarizes the framewise captions at fps 1 into a video one, where the central frame is handled by BLIP2 and the remaining are by tag2text.
> > >
> > > The motivation behind including coarse-level captions was based on the empirical evidence from the BLIP2 paper. This evidence demonstrated superior performance with image captioning benchmarks using BLIP2, albeit at the cost of longer inference speed. It was presumed that this advantage would extend to video-text contrastive learning as well.
> > >
> > > Given restricted compute resources, an ablation study was performed on two subsets of the dataset (InternVid-2M and InternVid-2M-BLIP), each having 2 million video-text pairs. InternVid-2M utilized fused captions, combining both coarse- and fine-level ones. In contrast, InternVid-2M-BLIP only used the coarse-level captions produced by BLIP2 on the central frames. For the mentioned using fine-level captions, concatenating the framewise captions from tag2text as the video captions is not a promising opinion as these captions are quite long and full of reptitions, unsuitable for contrastive learning. Thus, we did not include this setting in experiments.
> > >
> > > Zero-shot experiments were conducted on these models. Due to computational constraints, the ViCLIP-B was trained with a batch size of 4096 using 8 A100 GPUs, with a mask ratio set to 0.9. All remaining training parameters were consistent with those in the main paper.
> > >
> > > Contrasting the results from Table B and C, it's evident that the use of combined coarse and fine-level captions in video-text contrastive learning rendered superior zero-shot performance than utilizing the coarse level ones alone. It shows the effectiveness of our given video captioning pipeline.
> > >
> > > ### Table B. Zero-shot action recognition results of ViCLIP using different captions on Kinetics 400/600/700.
> > > |Method | Training Data | K400 |  | K600 |  | K700 |  |
> > > |:---:|:---:|:---:|:---:|:---:|:---:|:---:|:---:|
> > > | | |top-1 | AVG | top-1 | AVG | top-1 | AVG |
> > > |ViCLIP | +InternVid-2M | 51.70 | 64.69 | 49.20 | 62.34 | 40.90 | 53.70 |
> > > |ViCLIP | +InternVid-2M-BLIP2 | 38.40 | 51.58 | 36.40 | 49.19 | 29.10 | 40.68 |
> > >
> > > ### Table C. Results of zero-shot video retrieval from ViCLIP using different captions on MSR-VTT, LSMDC, DiDeMo, MSVD, and ActivityNet.
> > > |Method | Data | MSR-VTT | | LSMDC | | DiDeMo | | MSVD | | ANet | |
> > > |:---:|:---:|:---:|:---:|:---:|:---:|:---:|:---:|:---:|:---:|:---:|:---:|
> > > | | |T2V | V2T | T2V | V2T | T2V | V2T | T2V | V2T | T2V | V2T |
> > > |ViCLIP | +InternVid-2M | 31.8 | 33.7 | 14.3 | 12.7 | 13.6 | 21.5 | 39.6 | 62.5 | 9.9 | 16.8 |
> > > |ViCLIP | +InternVid-2M-BLIP2 | 21.7 | 21.9 | 5.2 | 5.1 | 7.1 | 12.7 | 24.6 | 42.1 | 6.4 | 10.2 |
> > >
> > > **Q2: I would appreciate more examples from the dataset, especially those that illustrate the contrast between coarse and fine-level captions.**
> > >
> > > We give two randomly picked examples to illustrate the difference between coarse and fine-level captions below. These examples highlight how coarse-level captions from BLIP2 provide a more detailed and comprehensive description of the scene or activity, as compared to fine-level framewise captions from tag2text. For instance, the BLIP2 caption in the second example identifies gender ("woman") and background context ("on a wooden floor"), which are details not included in the tag2text captions.
> > >
> > > - Case 1
> > >   - Fine-level framewise captions from tag2text: "a screenshot from a game of a pink van on the road at night", "a screenshot from a game of a pink van on the road at night", "a screenshot from a game of a pink van on the road at night", "a pink van on a small street at night", "a small truck on the road at night in the game", "a small van on the street at night", "a small truck on the road at night in the game", "a screenshot from a game of a pink van on the road at night"
> > >   - Coarse-level caption from blip2: "a small truck driving down the street in an online game"
> > > - Case 2:
> > >   - Fine-level framewise captions from tag2text: "a bowl of ice and a bowl of water", "a pair of hands mixing food in a bowl", "a pair of hands mixing food in a bowl", "a pair of hands mixing food in a bowl", "a person mixing a bowl of food with ice and water", "a person mixing food in a bowl of water", "a person mixing food in a bowl of water", "a person mixing food in a bowl of ice and water"
> > >   - Coarse-level caption from blip2: "woman making food on a wooden floor with bowls"

---

> > > > ### Author Response · Authors · 2023-11-20
> > > > **Response to Official Comment by Reviewer gWXX: Q3**
> > > >
> > > > **Q3: Regarding the T5-summary and Vicuna models mentioned, were they fine-tuned for processing captions? I suspect that these models are not capable enough for zero-shot handling of captions. Perhaps GPT-3.5-Turbo or GPT-4 might be suitable.**
> > > >
> > > > The [T5-summary](https://huggingface.co/mrm8488/flan-t5-large-finetuned-openai-summarize_from_feedback) and [Vicuna](https://huggingface.co/lmsys/vicuna-7b-v1.5) models we used have been fine-tuned on public NLP datasets for summary and dialogue performance respectively, but not specifically on our dataset for processing captions.
> > > >
> > > > It's true that larger language models like GPT-3.5-Turbo or GPT-4 could potentially improve the performance of caption handling due to their advanced capacity in capturing semantic meaning from text. However, these models also come with higher computational cost. Considering this trade-off between performance and efficiency, we elected to use a T5 summary model as it offers acceptable performance at a reasonable cost.
> > > >
> > > > Below are some qualitative comparisons between GPT-3.5-Turbo, GPT-4, and T5-summary illustrating their summary capabilities. These examples demonstrate that while the three models offer similar qualitative outcomes, choosing the best one would depend on specific requirements and constraints.
> > > >
> > > > - Case1:
> > > >   - Framewise Captions: "a woman with glasses talking to a man on the news", "a woman with glasses talking to a man on the news", "a woman with glasses talking to a man in an interview", "a woman with glasses talking to a man in a video", "a woman with glasses is talking to people in an interview", "a woman with glasses is talking to a man", "a woman with glasses talking to a man on the news", "a woman with glasses talking to a man on the news"
> > > >   - T5-Summary: "Woman with glasses talking to a man on the news."
> > > >   - GPT-3.5-Turbo: "A woman with glasses is speaking to a man in an interview or on the news."
> > > >   - GPT-4: "A woman with glasses is frequently seen talking to a man in interviews, videos, and on the news."
> > > >
> > > > - Case2:
> > > >   - Framewise Captions: "a screenshot of a ship in a large body of water", "a large ship in a large body of water", "a screenshot of a ship in a large body of water", "a large ship in a large body of water", "a screen shot of a boat in a large body of water", "a screen shot of a boat in a large body of water", "a screen shot of a boat in a large body of water", "a screen shot of a boat in a large body of water"
> > > >   - T5-Summary: "a screenshot of a boat in a large body of water"
> > > >   - GPT-3.5-Turbo: "a screenshot of a large ship or boat in a vast body of water"
> > > >   - GPT-4: "a large ship or boat in  a large body of water"

---

> > > > > ### Comment · Reviewer_gWXX · 2023-11-21
> > > > >
> > > > > Thanks for the author's detailed responses, most of my questions have been answered. Now I would like to see this paper accepted. I hope the author will include the rebuttal content in the final version.

---

> > > > > > ### Author Response · Authors · 2023-11-21
> > > > > > **Response to Official Comment by Reviewer gWXX**
> > > > > >
> > > > > > Thank you for your positive feedback and support towards the acceptance of our paper. We greatly appreciate the time and effort you put into the review process, as well as your constructive comments that have helped improve the quality of our work.
> > > > > >
> > > > > > Regarding your suggestion to include the rebuttal content, we completely agree with you. We will ensure all the important points addressed in the rebuttal are integrated into the final version of the paper.
> > > > > >
> > > > > > Once again, thank you for your insightful reviews and recommendations. This has been a valuable learning experience for us. We look forward to further collaborations in the future.

---

> ### Author Response · Authors · 2023-11-14
> **Response to Official Review of Submission4917 by Reviewer gWXX - Q3**
>
> **Q3: The performance of InterVid-10M-DIV, InterVid-10M-FLT is weird. It shows much better performance in zero-shot action recognition in Table 2, but poor in the fine-tuned setting of Table 3. The authors give reasons for false negatives, but what is the training batch size, given the much larger training data, I think the possibility of the same video clip appearing in the same batch is low.**
>
> We reason that the contrasting performance of ViCLIP (trained on InternVid-10M-FLT/DIV and InternVid-200M) in zero-shot vs. fine-tuned action recognition comes from differing task demands. Zero-shot performance largely hinges on aligning video and text representations, while fine-tuned action recognition focuses more on the discriminative abilities of the video representation. As such, there is no guaranteed correlation between a model's capabilities in these two tasks.
>
> Regarding your observation about InternVid-10M-DIV/FLT, we agree it performs better in zero-shot action recognition (Table 2), yet lacks in the fine-tuned setting (Table 3). Our explanation is that ViCLIP trained on InternVid-200M has a more robust video representation due to a higher quantity and diversity of videos. Conversely, the version trained on InternVid-10M-FLT/DIV likely excels in video-text alignment: a result of reducing the number of clips sourced from identical videos. This approach makes their training data more distinct than InternVid-200M’s, which promotes contrastive learning to enhance video-text representation—yielding superior outcomes for zero-shot action recognition and video-text tasks such as retrieval.
>
> In response to the query about false negatives and batch sizes, you're right that the same video clip appearing in the same batch is relatively rare given the extensive size of the training data. However, we consider an alternative explanation for the false negatives: the higher video diversity in InternVid-10M-FLT/DIV and its improved video-text correlation (sorted by CLIP) result in a more effective video-text representation compared to InternVid-10M.

---

### Meta-Review · Area_Chair_MqG5 · 2023-12-05

**Metareview:**

The paper introduces a large-scale video-text dataset, InternVid, encompassing 234M video-text pairs spanning 760K hours, aimed at enhancing video representation learning. This dataset addresses the challenges of noise and low correlation in ASR transcripts by employing captioning models and a pre-trained language model for generating accurate video descriptions. The authors also present a baseline model, ViCLIP, pre-trained on this dataset, which has strong performance in tasks such as action recognition and text-video retrieval.

The reviewers found this paper well written and useful to the ICLR community and having this data could represent the next step for improved video understanding. I will recommend acceptance and I suggest the authors incorporate all the obtained feedback.

**Justification For Why Not Higher Score:**

- Borderline, tending towards spotlight

**Justification For Why Not Lower Score:**

- Highly relevant for our image-(over)saturated field.
- Well executed, potential to bring video understanding to the next level.

---

### Decision · Program_Chairs · 2024-01-16

Accept (spotlight)